# The origins of the Guinness stout yeast

Daniel W. M. Kerruish [1✉], Paul Cormican[2], Elaine M. Kenny [2], Jessica Kearns[1], Eibhlin Colgan[1], Chris A. Boulton[3] & Sandra N. E. Stelma[1]

Beer is made via the fermentation of an aqueous extract predominantly composed of malted barley flavoured with hops. The transforming microorganism is typically a single strain of *Saccharomyces cerevisiae*, and for the majority of major beer brands the yeast strain is a unique component. The present yeast used to make Guinness stout brewed in Dublin, Ireland, can be traced back to 1903, but its origins are unknown. To that end, we used Illumina and Nanopore sequencing to generate whole-genome sequencing data for a total of 22 *S. cerevisiae* yeast strains: 16 from the Guinness collection and 6 other historical Irish brewing. The origins of the Guinness yeast were determined with a SNP-based analysis, demonstrating that the Guinness strains occupy a distinct group separate from other historical Irish brewing yeasts. Assessment of chromosome number, copy number variation and phenotypic evaluation of key brewing attributes established Guinness yeast-specific SNPs but no specific chromosomal amplifications. Our analysis also demonstrated the effects of yeast storage on phylogeny. Altogether, our results suggest that the Guinness yeast used today is related to the first deposited Guinness yeast; the 1903 Watling Laboratory Guinness yeast.

[1] Diageo Ireland, St James's Gate, The Liberties, Dublin, Ireland. [2] ELDA biotech, Kildare, Ireland. [3] Brewing Consultant, Burton on Trent, UK. ✉email: daniel.kerruish@diageo.com

To mitigate against unpredictable climate and food shortages humans moved from hunter gathering to an agrarian existence[1]. In consequence, animals, plants and unwittingly, some microorganisms were domesticated[2] including the ethanol-forming and $CO_2$-generating yeast *S. cerevisiae*. The latter event occurred in multiple geographical locations resulting in the almost universal exploitation of the abilities of *Saccharomyces* yeasts to leaven bread and produce a multitude of alcoholic beverages[3,4].

Largely based on its ease of cultivation and *GRAS* designation *S. cerevisiae* has become the model for eukaryotic cell biology[5,6]. Principal component analysis of single nucleotide polymorphisms (SNPs) have established a Chinese origin[3] of *S. cerevisiae* with the *Saccharomyces* genus believed to be of Asian origin[7]. Publications using industrial strains have provided further understanding of the phylogenetic origins of *S. cerevisiae* used in wine and beer production[4,8]. Recent publications have established the effects of European and Asian wine admixture on today's industrial brewing yeast[9] with those yeast more readily associated with commercial brewing activities dominated by Asian admixture[10]. Admixture within domesticated *S. cerevisiae* is incongruous with other *Saccharomyces* yeast, where admixture and heterozygosity levels are low[7]. Many of these industrial *S. cerevisiae* strains have genomes which feature polyploidy, aneuploidy and loss of fitness, all characteristic of domestication[3]. These domesticated yeasts carry mutations conferring properties that make them suitable for beer fermentations. These include the formation of a spectrum of beer flavour yeast metabolites, the ability to utilise sugars such as maltotriose and maltose; ethanol tolerance and an ability to separate from beer at the end of fermentation via the process of flocculation[8].

Historically, distinct beer styles were associated with specific geographical regions based on the availability of raw materials and the mineral composition of the water supply[11]. In addition, specific yeast strains were selected to enhance and meet the desired beer style qualities. Initially this was unwitting since the role of yeast was only proven in the early 19th century independently in France by Cagniard-Latour and Germany by Schwann and Kutzing[11]. Nevertheless, brewers had actively been selecting yeast for their phenotypes in Europe from at least the 1600s[4]. This led to selection of specific industrial brewing yeast strains from geographical locations containing specific SNPs. Confirmation of geographical specific SNPs have been observed in yeasts isolated from beers brewed in Germany, Belgium, Britain, the USA and Norway[4,12].

Founded in 1759 by Arthur Guinness, the St James's Gate Dublin brewery was by 1886 the world's largest by volume[13] (1.8 MHl per annum) becoming synonymous with stout beer brewed in the Irish dry stout style. The brewery archives provide comprehensive details of the yeast studies performed by scientists employed by Guinness; however, no information is held regarding the origins of the yeast first used to brew beer at St James's Gate. The first mention of a starting culture (yeast) is detailed in the 1809 Guinness Brewing Book on the 10th and 11th of May. At the time of writing, the two principal Guinness stouts; Guinness Irish Draught Stout (IDS) and Guinness Foreign Extra Stout (FES) are brewed using yeast isolated from the 1959 Guinness pitching yeast. The Guinness archives confirm that the 1959 isolates were selected from the 1903 Watling Laboratory Guinness yeast.

In this study, 13 Guinness yeast strains, the two current production yeasts and six other historical Guinness strains were assessed to establish the origins and nature of the Guinness yeast. Our analyses have established that the Guinness yeast form a subgrouping within the previously described Beer 1 clade[4]. Furthermore, the Guinness strains are mosaic sharing ancestral lineage that is different to other historical Irish brewing yeast.

Genomic assessment of the Guinness yeasts confirmed that the Guinness yeast are genetically similar with different phenotypes, demonstrating the importance of phenotypic validation of yeast for brewing. The assessment of the Guinness yeast chromosome and copy number variation (CNVs) supports previous conclusions regarding the role of gene copy number and phenotype. Consequently, the principal findings of this paper are that the Guinness yeast strains are intimately related and are derived from a common ancestor.

## Results

**Determining the origins of the Guinness yeast.** The two current Guinness production and eleven historical Guinness yeast were chosen to understand the origins and phylogeny of Guinness yeast. Selection was based on a review of historical records held in the Guinness archives (Table 1). The purity of the 13 Guinness yeast strains was determined using the interdelta PCR method of the TY retrotransposon elements[14]. The method generated a 'fingerprint' for each of the yeast strains (Supplementary Fig. 1). Dendrograms of the PCR fragment lengths were assessed using hierarchical clustering (Euclidean distance) with yeast confirmed as being identical given a 100% similarity score. For the two Guinness production yeast, FES and IDS, the interdelta PCR method established that all three FES strains were <100% similar, two of IDS strains were 100% similar and the third 98%. Interdelta PCR assessment of the other eleven historical Guinness yeast established the principal fingerprint of the strains; subsequently representative samples of each of the eleven historical Guinness yeast were determined. The fingerprinting evaluation of the 13 Guinness yeast, two production yeast encompassing the five isolated individual yeast strains, and the eleven historical yeasts, resulted in a selection of 16 Guinness yeast for whole-genome sequencing.

In addition to the 16 selected Guinness yeast a further 6 historical Irish brewing yeast (Table 1) were sequenced using an Illumina MiSeq machine reading with a minimum depth of 30× coverage using 2 × 250 bp paired reads. The Sequences were assembled de novo using the *S. cerevisiae* S288C reference genome (R64 1-1). Origins of the Guinness strains were probed by comparing their genomes with those of the 6 other Irish Brewing yeasts and 154 previously published *S. cerevisiae* strains[4]. A total of 466,327 filtered variant sites were identified: 434,890 SNPs and 31,427 indels. For the Guinness yeast, 96,821 filtered variant sites were identified with: 88,271 SNPs and 8550 indels (Table 2). Some 5407 filtered variant sites were exclusive to the Guinness yeasts, a total of 4907 SNPs and 500 indels.

A maximum-likelihood (ML) phylogenetic tree of the 176 yeast was constructed using RAxML v8.2.12[15] based on the concatenated alignment of orthologues protein coding genes and visualised using ggtree (v 3.6.2)[16] (Fig. 1). Previous work has placed the 154 *S. cerevisiae* strains into 8 separate lineages[4]. Brewing strains were located in either the Beer 1 or Beer 2 lineage. Our analysis confirmed these observations with the Guinness and other historical Irish brewing yeast placed within the Beer 1 clade (Fig. 1).

The Beer 1 clade contains three separate subpopulations; a consequence of allopatric activity. These three distinct geographical groupings are Belgium/Germany, Britain and the USA. The non-Guinness Irish brewing strains were located in the 'Britain' subpopulations, whereas the Guinness yeast occupy their own subgroup outside the USA and 'Britain' subpopulations. This was an unexpected observation as the Guinness archives contain multiple records of the company requesting and supplying yeast to other Irish brewers, examples of a practice common until the mid-20th Century.

**Table 1 Name and description of yeast strains used in this study and selected following a literature review of the Guinness archives.**

| Yeast | Name | Source | Brewing group |
|---|---|---|---|
| Irish Draught Stout 1* | IDS1 | 1959 St James's Gate Pitching Yeast | Guinness |
| Irish Draught Stout 2* | IDS2 | 1959 St James's Gate Pitching Yeast | Guinness |
| Foreign Extra Stout 1* | FES 1 | Class I mutant from IDS | Guinness |
| Foreign Extra Stout 2* | FES 2 | Class I mutant from IDS | Guinness |
| Foreign Extra Stout 3* | FES 3 | Class I mutant from IDS | Guinness |
| 1947 | 1947 | 1947 St James's Gate Pitching Yeast | Guinness |
| 1950 | 1950 | 1950 St James's Gate Pitching Yeast | Guinness |
| 1955 | 1955 | 1955 St James's Gate Pitching Yeast Pitching yeast | Guinness |
| 1959 brewing yeast | 59 BY | Co-flocculant yeast used with IDS between 1959 and 1963 to aid beer processing | Guinness |
| 1959 | 1959 | 1959 St James's Gate Pitching Yeast | Guinness |
| Ikeja | Ikeja | Yeast selected from the 1959 St James's Gate Pitching Yeast. Used in the first Guinness African Brewery, Ikeja, Lagos, Nigeria | Guinness |
| 1960 | 1960 | Yeast reselected from 1947 St James's Gate Pitching Yeast | Guinness |
| 1981 | 1981 | Yeast reselected from IDS | Guinness |
| Park Royal 1960 | PR1960 | 1960 Guinness Park Royal Brewery Pitching Yeast | Guinness |
| Park Royal 1979 | PR1979 | 1979 Guinness Park Royal Brewery Pitching Yeast | Guinness |
| Park Royal 1986 | PR1986 | 1986 Guinness Park Royal Brewery Pitching Yeast | Guinness |
| Cherry | Cherry 1960 | Cherry's Pitching Yeast | Cherry |
| Great Northern Brewery 1958 | GNB 1958 | Great Northern Brewery Stout Pitching Yeast | GNB |
| Macardle 1965 | Macardle 1965 | 1965 Macardle Pitching Yeast | Macardle Moore |
| Macardle 1993 | Macardle 1993 | 1970 Smithwick's Pitching latterly used as Macardles Pitching Yeast | Macardle Moore |
| Perry | Perry 1967 | 1967 Perry Pitching Yeast | Perry |
| Smithwicks* | Smithwicks 1986 | 1986 Smithwicks Production Yeast | Smithwick's |

*Denotes yeast presently used in beer production.

**Table 2 Sporulation percentage; mean sequencing coverage along *S. cerevisiae* S288C genome, transition, transversion and singleton SNPs, and total indels of the 16 sequenced Guinness yeast.**

| Guinness yeast | Sporulation (%) | Average sequencing coverage (x) | Total transition SNPs | Total transversion SNPs | Number of singletons | Total indels |
|---|---|---|---|---|---|---|
| IDS1 | 0 | 417.6 | 59787 | 20606 | 142 | 7033 |
| IDS2 | 0.03 | 1230.1 | 59226 | 20494 | 56 | 7111 |
| FES 1 | 1.09 | 391.8 | 59740 | 20579 | 168 | 7075 |
| FES 2 | 0.13 | 713.4 | 59841 | 20659 | 30 | 7099 |
| FES 3 | 0 | 968.1 | 60000 | 20715 | 29 | 7166 |
| 1947 | 0.2 | 835.8 | 58899 | 20322 | 59 | 7073 |
| 1950 | 0 | 770.5 | 58795 | 20238 | 39 | 7017 |
| 1955 | 0 | 767.8 | 60012 | 20731 | 1676 | 7198 |
| 59 brewing yeast | 0 | 879.2 | 58392 | 20082 | 84 | 6855 |
| 1959 | 2.77 | 996.3 | 58525 | 20163 | 93 | 6999 |
| Ikeja | 0.79 | 1010.1 | 59703 | 20618 | 334 | 7079 |
| 1960 | 0.1 | 849.9 | 57754 | 19935 | 303 | 6897 |
| 1981 | 0.2 | 531.2 | 60227 | 20815 | 117 | 7181 |
| Park Royal 1960 | 0.03 | 888.1 | 57735 | 19978 | 342 | 6885 |
| Park Royal 1979 | 0 | 718.7 | 59815 | 20637 | 84 | 7138 |
| Park Royal 1986 | 0 | 304.6 | 59898 | 20694 | 132 | 7057 |

Sporulation percentage was determined using the ASBC sporulation method[93]. Sequencing analysis of the Guinness yeast were performed using an Illumina MiSeq machine reading with a minimum depth of 30× coverage using 2 × 250 bp paired reads. SNPs and indels were determined through de novo assembly of the Guinness yeast compared to the reference yeast *S. cerevisiae* S288C.

To understand the origins of the Guinness yeast populations, structure and degree of admixture were determined for the 176 genomes used in this study using FastSTRUCTURE (Version 1.0)[17]. Owing to the high degree of sequence similarity between the Guinness samples a single representative, IDS1, was selected for analysis. Varying the number of ancestral populations (K) between 1 and 10, $K = 8$ was found to be optimal (Fig. 1); an observation in accordance with previous ancestral population investigations[4]. The non-Guinness Irish brewing strains were all placed within the 'Britain' subpopulation, based on >80%

common ancestry. The brewing strains of companies Perry, Cherry and Smithwicks aligned completely to the Britain subpopulation, whereas the Great Northern, Macardles 1966, and Macardles 1993 yeast aligned with the 'Britain' group (>88%) but also the US and Belgium/Germany subpopulations.

Mosaicism within the phylogenetic tree is defined as a yeast possessing an ancestry of <80% from a single population[4]. Within the Beer 1 subpopulation, 10 yeasts were designated as being mosaic; these yeasts that are marked in highlighted orange in Fig. 1. Analysis of the representative Guinness yeast, IDS1,

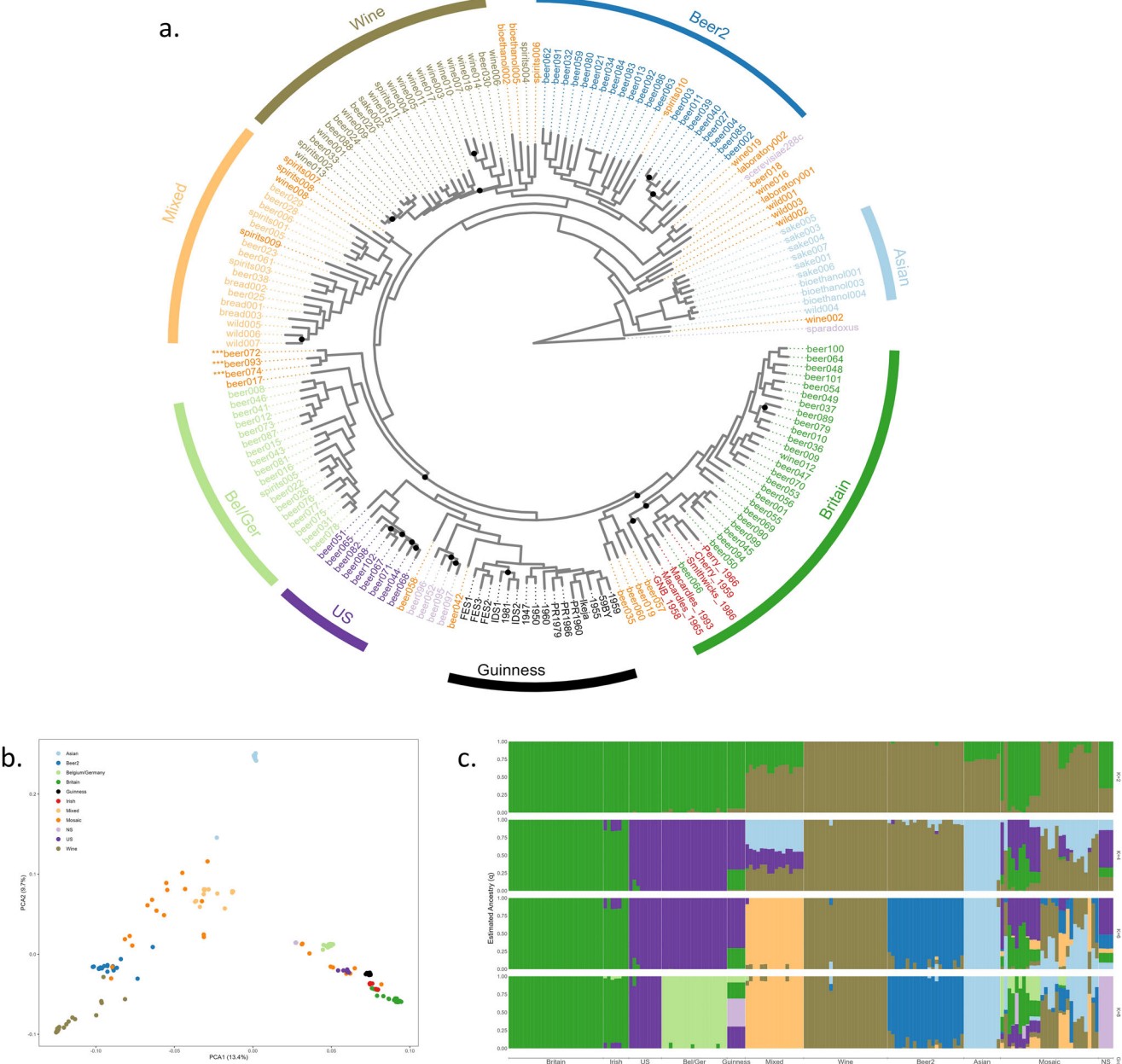

**Fig. 1 Phylogeny and population structure of the Guinness yeast and other industrial *S. cerevisiae* strains. a** Guinness and other Irish brewing yeast within the maximum-likelihood phylogenetic tree of *S. cerevisiae*. Guinness and other Irish brewing yeast were sequenced using an Illumina MiSeq platform and combined with 154 previously sequenced *S. cerevisiae*[4]. Branch length reflects the number of substitutions per site, with colour denoting the yeast lineage. A maximum-likelihood (ML) phylogenetic tree was reconstructed in RAxML v8.2.4[15], performing 100 iterations to search for the best tree, using a discrete GTRGAMMA model of rate heterogeneity. Bootstrap branch support was assessed by performing 1000 pseudoreplicates. Trees were visualised using ggtree (v 3.6.2)[16]. Yeasts that are marked in highlighted orange are described as being Mosaic[4]. Yeast marked with three asterisk are used to brew beers in the Hefeweizen style. **b** Principal component analysis of 434,890 SNPs sites from the assessed 176 *S. cerevisiae* strains. Population differences indicated by colour; NS not specified. **c** Population structure of the 434,890 SNPs sites of the *S. cerevisiae* strains used in this study. IDS1 was used as a representative Guinness yeast sample for all Guinness yeast due to the high degree of sequence similarity analysis consequently 161 genomes admixture were assessed. Resolved population fractions are represented by the vertical axis; colours denote estimated ancestral membership. Varying the number of ancestral populations (K) between 1 and 10 using the simple prior implemented in fastSTRUCTURE[17], $K = 8$ found to be optimal.

established the ancestry of IDS1 being <80% from one grouping. The Guinness yeast are therefore Mosaic. Further analysis of the admixture (Supplementary Fig. 2) using the Alpaca[18] software confirmed the observations of the FastSTRUCTURE[17] analysis, with the genome hereditary of the Guinness yeast belonging to multiple ancestry origins. The admixture and Alpaca[18] analysis establish a non-linear monophyletic origin of the Guinness

strains. The non-overlapping segments across the 16 chromosomes display a high degree of similarity with segments from yeasts derived from very distinct phylogenetic subclades and geographical locations. Consequently, analysis present in Supplementary Fig. 2 confirmed the highly non-linear genetic content of the Guinness yeast relative to other members of the Beer 1 clade. Furthermore, the phylogenetic tree presented in Fig. 1 placed the

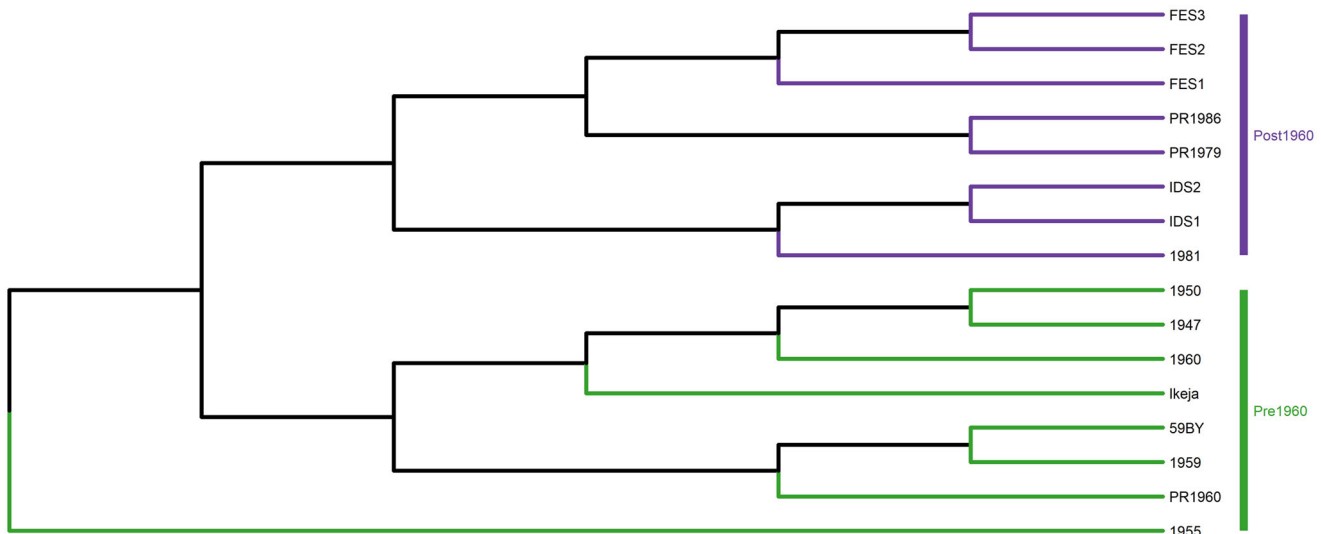

**Fig. 2** Hierarchical clustering of the 16 Guinness yeast using standard dissimilarity matrix of 20,000 protein coding biallelic SNPs as determined by de novo assembly of the Guinness yeast to the MinION sequenced reference genome IDS1.

yeast Beer042 as the closest relative to the Guinness yeast. The Alpaca[18] analysis confirms this observation with all of the chromosomes sharing SNPs observed in Beer042. Omission of the Beer042 from the analysis confirmed that SNPs from the 'not specified' ancestry were the most significant (38.8%) (Supplementary Fig. 2).

**Phylogeny assessment of the Guinness yeast using long read sequencing technology.** Short read sequencing de novo assembly against a reference genome introduces potential scaffolding bias[19]. It has been observed that using *S. cerevisiae* S288C as the reference genome results in large sequencing gaps in sub-telomeric regions and in consequence poor yeast genome assembly[20]. Long read sequencing provides a more comprehensive genome assessment from which a reference genome can be assembled and annotated[20]. Using a Guinness yeast as the reference genome is suitable since *S. cerevisiae* S288C is a distant relation and this could result in potential phylogeny misrepresentation. The Guinness yeast IDS1 was selected for long read sequencing using the Minion system (Oxford Nanopore Technology, Oxford, UK). 421,040 reads were generated and the genome assembled using Flye[21] (v 2.9) programme. The other 15 Guinness yeast were then assembled against IDS1. Following removal of intergenic variants, a total of 20,039 SNPs present in protein coding genes were identified in the 16 Guinness strains. Hierarchical clustering of the 16 Guinness strains was established using standard dissimilarity matrix analysis. The analysis of all SNPs identified in protein coding genes resulted in assessment of 20,000 protein coding biallelic SNPs[3] (Fig. 2). With the exception of the 1955 Brewing Pitching yeast the Guinness yeast divide into two groupings, pre and post-1959. The observed hierarchical clustering generated using IDS1 as the reference genome was different from the observed clustering derived using *S. cerevisiae* S288C. Analysis using the former indicated that there was a deliberate selection of the Guinness yeast in 1959 confirming the archival account of Robert Gilliland's 1959 Guinness yeast reselection programme.

**Copy number variation and chromosomal arrangement of the Guinness yeast.** To determine the CNV for the different Guinness strains a full normalised read depth analysis was performed in 250 bp windows across each chromosome. The result was

normalised against an estimated copy number of 4 as this was the average chromosome copy number estimated for the Guinness strains. The likely accuracy of Guinness ploidy estimates was confirmed by reassessing the ploidy of previously published yeast samples[4]. The 250 bp window assessment of the Guinness strains (Fig. 3) showed the presence of multiple copies of chromosomes and CNVs within individual chromosomes. Whilst there are chromosomal CNVs across all 16 chromosomes only chromosomes II, XIII and XV have 5 chromosomal copies in 6 or more Guinness strains; even yeast that are closely related, IDS1 and IDS2, exhibit difference in CNV in chromosome II, VI, X and XVI. Consequently all 16 Guinness yeast are aneuploid.

Aneuploidy is common for ale brewing strains and has been previously reported in multiple studies[4,11,22]. Unlike natural isolates, domesticated yeast have been influenced by human activity. This selection pressures result in polyploidy for genes conferring sought-after phenotypic qualities. In concert with this, decreased global cellular fitness occurs[3]. Consequently, wild type *S. cerevisiae* are more likely to be diploid with a functioning sexual phenotype[22]. In contrast and presumably a consequence of the effects of aneuploidy, all 16 of the Guinness strains exhibited poor sporulation (Table 2); 7 strains did not sporulate at all and of the other 9 the 1959 pitching yeast had the highest sporulation percentage of just 2.8%. This observation is concurrent with Bilinski and Casey[23] who also reported poor sporulation in aneuploid yeast.

**Phenotype analysis of the Guinness yeast.** In order to determine whether the genetic similarity defined a 'Guinness yeast phenotype', the 16 strains were assessed for phenotypic traits using mini-fermentations as described in the methods section. Some differences in both patterns and extent of attenuation, as well as ethanol production were observed (Fig. 4). A one way ANOVA with Tukey's post-hoc test of the 16 Guinness yeast determined that ethanol production was significantly different ($P = 3.09 \times 10^{-9}$). Assessment of all of the Guinness yeast using a single representative IDS, FES and Park Royal yeast confirms that ethanol production is significantly different (one way ANOVA $P = 0.00036$), but separate one way ANOVA analysis of the ethanol production of the two yeast identified within the IDS yeast were not statistically significant ($P = 0.64$) likewise analysis of the Park Royal yeast showed that there was no statistical

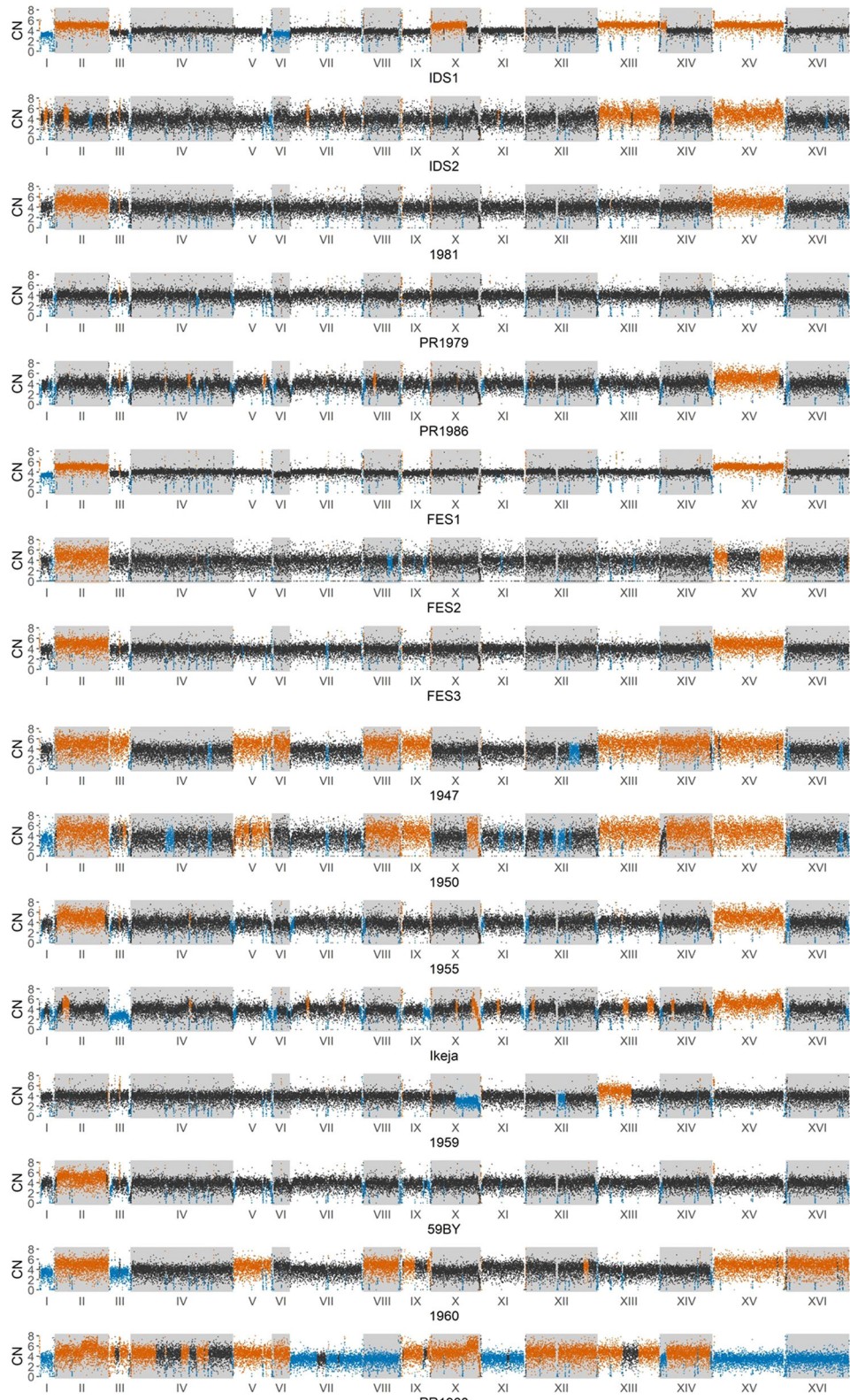

**Fig. 3 Estimated CNV in 250 base pair non-overlapping windows across the entire genome of the 16 Guinness yeast.** A black dot on a plot represents a window where the estimate copy number is 4. A blue dot represents a region with an estimated loss of copy number (<4) and an orange dot represents a region of estimated increased copy number (>4).

difference between the Park Royal yeast ($P = 0.13$). Only the FES yeast demonstrated significant difference in ethanol production (One way ANOVA $P = 0.00011$) but when FES 2 and FES 3 were assessed without the inclusion of FES 1 ethanol production was determined not to be statistically significant ($P = 0.08$). Likewise, time to attenuation varied between strains with the Park Royal 1979 yeast achieving attenuation in 22 h compared with 72 h for the 1959 Guinness pitching yeast (Fig. 4).

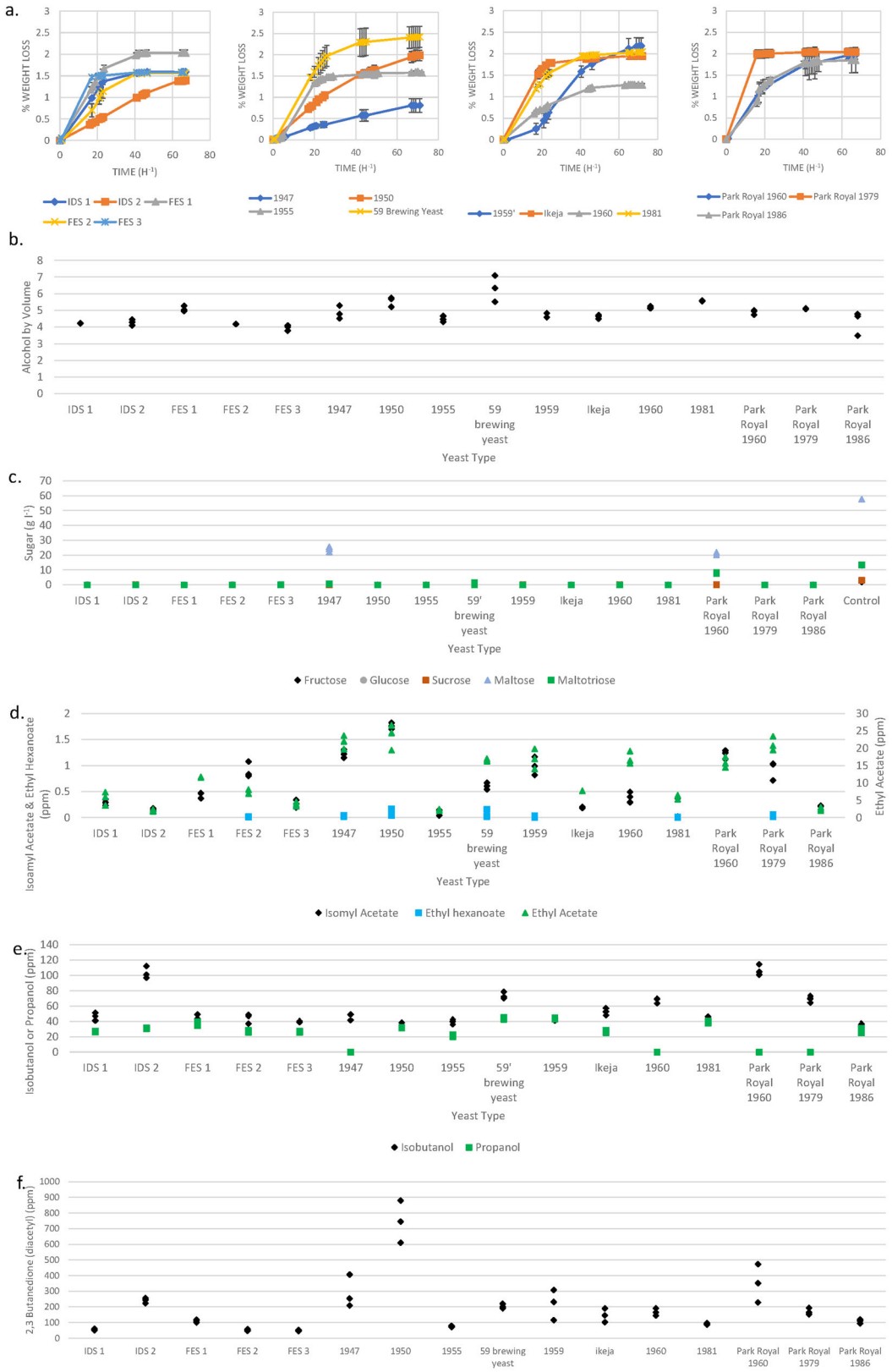

**Fig. 4 Phenotypic assessment of the Guinness yeast. a** Percentage weight loss, **b** ethanol production, **c** sugar concentration **d** assessment of esters, **e** higher alcohols and **f** 2, 3 butanedione (diacetyl) production at the cessation of fermentation using the different Guinness yeast. Fermentations were performed using 100 ml of 12ºP all-malt wort, with an inoculation rate of $1 \times 10^7$ ml$^{-1}$ cells. Samples were incubated at 25 °C and stirred at 250 rpm. Observations presented are $n = 3$ biologically independent experiments. In **a** each time point represents the mean of 3 independent replicates and standard errors are shown as SD ± error bars.

The progress of brewing fermentations is typically assessed by recording the decrease in wort density. Measuring loss of weight, in the mini-fermentations described here is an indirect method which also relies on a fall in density. This decrease is a consequence of the utilisation of wort sugars by yeast and the formation of ethanol. Brewing yeast strains can assimilate simple wort sugars which include glucose, fructose, sucrose, maltose and maltotriose; dextrins are not fermented[24].

Maltotriose utilisation correlates with the maltose multigene loci, five of which (MAL1, 2, 3, 4 and 6) have been identified in S. cerevisiae[25]. Only the MAL1 and MAL3 multigene are present within the reference yeast S. cerevisiae S288C. The MAL locus comprises three genes, a maltose permease (gene 1), maltase (gene 2) and a Trans acting MAL-activator (gene 3). Illumina sequencing of the 16 Guinness strains confirmed the presence of MAL1 and MAL3. A homozygous premature stop codon was identified in the MAL1 maltose permease gene (MAL11) for all 16 Guinness yeast (Supplementary data 1). This stop codon mutation potentially prevents the loss of gene function, although the occurrence of this stop codon is present within 145 of the 176 yeast assessed in this study and maltotriose is utilised by all of the Guinness yeast (Fig. 4). Within the phylogenetic tree disruption to MAL11, whilst present in a small number of Beer 1 yeast, is more readily associated with brewing yeast present in the Beer 2 brewing yeast grouping[4].

Illumina analysis of MAL3 of S288C, established the presence of 16 heterozygous frameshift mutations in the maltose permease, MAL31. Similar mutations were present in all the different Guinness yeast strains (Supplementary Data 1). The frameshift mutations in MAL31 did not result in loss of maltose and maltotriose utilisation as assessment of maltose and maltotriose at the end of the fermentation confirmed the consumption of these wort sugars (Fig. 4).

Further analysis of the Guinness strains using the longer nanopore sequencing reads determined that the MAL6 locus was present. When detected in other S. cerevisiae strains the MAL6 multigene locus is located on Chromosome VIII and is arranged from the centromere as MAL63, MAL61 and MAL62[26]. In contrast, assessment of the Guinness strains showed that MAL61 and MAL62 were arranged on Chromosome VIII, as expected; however, MAL63 mapped to the sub-telomeric region of chromosome XVI. In the reference strain S. cerevisiae S288C this Open Reading Frame (ORF) on chromosome XVI is designated as gene YPR196W and is described as a "Putative maltose-responsive transcription factor". The unusual arrangement of the MAL6 locus appears to be specific to the Guinness yeast and should be considered provisional until confirmed through additional experimental data.

**Production of flavour metabolites by Guinness yeast**. Beer recovered from the completed mini-fermentations were analysed for the flavour active esters: isoamyl acetate, ethyl butyrate, ethyl hexanoate and ethyl acetate (Fig. 4). All were detected in the Guinness yeast fermentations apart from ethyl butyrate. Ethyl hexanoate was produced by 7 of the 16 Guinness strains, but at concentrations lower than the flavour threshold of 0.2 ppm[27]. The concentrations of ethyl esters varied with strain. In the case of ethyl hexanoate, isoamyl acetate and ethyl acetate the differences were statistically significant; more so in the case of the latter two esters, one way ANOVA $p = 0.011$, $p = 9.32 \times 10^{-20}$ and $p = 6.91 \times 10^{-20}$, respectively. The 1947, 1950 and Park Royal 1960 yeasts produce isoamyl acetate at concentrations above the flavour threshold (1.1 ppm[27]) whereas ethyl acetate is the most widely produced ethyl ester. Eight of the 16 Guinness yeast strains produce ethyl acetate above the flavour threshold of 10 ppm[27]. A

one way ANOVA of these 8 yeast confirms that ethyl acetate production in the Guinness yeast is strain-specific ($p = 8.06 \times 10^{-5}$); however, further analysis of ethyl acetate production of the four Guinness yeast collected in 1959 and 1960 show no statistical significance (One way ANOVA $p = 0.86$), likewise there is no statistical significance between the 1947 and 1950 pitching yeast (One way ANOVA $p = 0.51$).

Higher alcohols (fusel alcohols) are the most abundant yeast-derived organoleptic compounds present in beer apart from ethanol[28]. Isobutanol and propanol impart solvent and alcohol/sweet aromas to the beer. As in the case of ethanol production, the concentrations of higher alcohols arising in the beers varied with strain (Fig. 4); one way ANOVA isobutanol $p = 4.79 \times 10^{-21}$ and propanol $p = 4.42 \times 10^{-32}$.

Higher alcohols are formed by transamination, decarboxylation and reduction via the Ehrlich pathway[24]. Transamination is rate-determining and over-expression of BAT2 results in increased higher alcohol production[29,30]. CNV of the genes responsible for the Ehrlich pathway (Supplementary data 2) established that CNV of BAT2 differed between the Guinness strains with a median value of 4. The Park Royal 1960 produced the highest concentration of isobutanol and was shown to have 6 copies of the BAT2 gene. The Guinness production yeast IDS2 produce similar amounts of isobutanol compared to the Park Royal 1960 strain ($t$-test $P = 0.61$) and has 4 copies of BAT2. Both IDS1 and IDS2, have the same CNV of BAT2, but IDS1 produced significantly less isobutanol ($t$-test $P = 0.00048$) indicating the involvement of other factors. In fact, a frameshift mutation was identified in the THI3 decarboxylase gene and the dehydrogenase AAD6 gene was not detected in the Guinness yeasts; deletions of THI3 and AAD6 have been observed to negatively affect isobutanol production[30].

The vicinal diketone, diacetyl (2,3 butanedione) arises in beer during fermentation where it imparts a buttery or butterscotch like flavour[31]. Diacetyl is formed indirectly by brewing yeast during fermentation from α-acetolactate. The latter is an intermediate in the isoleucine valine (ILV) synthetic pathway and part of the pool is exported from yeast cells where it undergoes spontaneous oxidative decarboxylation in fermenting wort to form diacetyl. Diacetyl was present at the end of fermentation for all of the Guinness yeast (Fig. 4); however, only 7 of the 16 strains at a concentration greater than the flavour threshold of 100–400 ppb[32]. Differences in diacetyl concentrations for individual strains were highly statistically significant (one way ANOVA $p = 7.09 \times 10^{-13}$).

The diacetyl precursor α-acetolactate is produced by the enzyme acetolactate synthase[32]. The responsible genes ILV2 and ILV6[33,34] were found in all Guinness yeasts, with a total of 11 SNPs present in ILV2 compared to the reference yeast S. cerevisiae S288C. A total of 5 SNPs, and two non-synonymous mutations, for ILV6 were found in all of the Guinness strains (Supplementary data 3). 4 of the 5 Guinness yeasts which produced the highest residual diacetyl concentration had 5 copies of the ILV2 gene compared to a median value of 4 CNV. The apparent correlation between copy number of ILV2 and residual diacetyl concentration could be causative; for many traditional beers' diacetyl removal occurs via lengthy periods of storage, post-fermentation, at cool temperatures in the presence of yeast. For the Guinness yeast it was observed that the strains with the most rapid fermentations and corresponding longer post-fermentation time had the lowest diacetyl concentration (Fig. 4). This is not surprising since diacetyl is reduced principally in late fermentation through passive uptake by yeast and subsequently enzymatic conversion of diacetyl first to acetoin and thence to 2,3-butanediol[35,36]. Consequently for FES production, where the presence of diacetyl is part of the beer's flavour

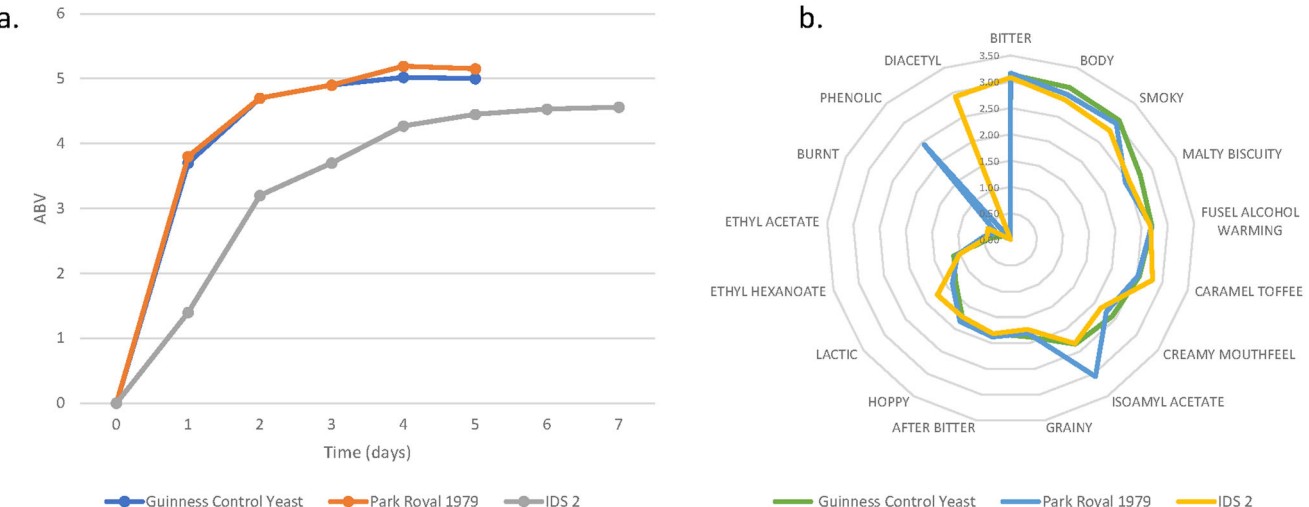

**Fig. 5 1HL fermentation and flavour assessment of the Guinness Park Royal 1979 and IDS2 yeast strains compared to present Guinness production yeast. a** Rate of fermentation and **b** flavour of Guinness Irish Draught Stout brewed using a control Guinness yeast from Dublin St James's Gate and the Guinness yeasts: Park Royal 1979 and IDS2. All fermentations were conducted in 100 L fermentation vessels with Guinness wort collected from St James's Gate Brewery. The tasting samples were assessed, in duplicate using the Guinness Draught attribute list by an expert panel using Quantitative Descriptive methodology. A minimum of $n = 18$ assessors were used to determine flavour attributes.

profile, reducing yeast contact time post attenuation ensures that the diacetyl flavour remains within the beer.

Fermentations were repeated at a scale of 1hl as fermentations at this increased scale are more representative of commercial-scale brewing. Additionally, the increased volume allowed for the resultant beers to be subjected to standard sensory profiling. Fermentations were carried out in a pilot brewery using 12-degree Plato (ºP) wort. Degree Plato is a measurement, related to density, and used by brewers to determine the concentration of dissolved solids including fermentable and non-fermentable sugars in brewers' wort. Guinness stout wort was sourced from St James's Gate Brewery, Dublin. Two Guinness strains, Park Royal 1979 and IDS2 were chosen for the trial as the Park Royal 1979 time to attenuation was the shortest of the 16 Guinness strains and the IDS2 yeast was chosen as it was the atypical IDS production yeast. The chosen yeasts were compared with a control, a third-generation production Irish Draught yeast culture taken from the St James's Gate brewery. Third generation refers to a culture that had already been used for 3 previous cycles of serial fermentations with intermediate cropping and storage. Yeast cultures of this "age" is considered to produce standard fermentation performance and generate typical beer. When the fermentations were completed, the resultant beers were processed using the standard Irish Guinness stout procedure. Beers were assessed chemically via analysis and organoleptically via the Guinness external taste panel using quantitative descriptive methodology.

The results were largely in accord with those obtained from the mini-fermentation study (Fig. 5). The flavour panel detected isoamyl acetate and phenolic off-flavour in the beer made with Park Royal 1979, and diacetyl in Guinness made using IDS2; these observations were confirmed by GC-MS analysis (Supplementary Fig. 3). Times to attenuation and final ethanol concentrations for both small and larger scale fermentations were also similar.

**Phenolic off-flavour (POF) phenotype of the Guinness yeast.** The formation of 4-vinyl guaiacol, also known as *Phenolic off-flavour* (POF), imbues beers with a medicinal, clove-like aroma and flavour. It is produced from a precursor, ferulic acid, derived

from cereal grains, via expression of yeast genes[37]. All the Guinness yeast strains used in this study were POF⁺ (Fig. 6). The degree of POF character which developed varied with yeast strain (one way ANOVA P value = $4.7 \times 10^{-21}$). The flavour threshold of 4-vinyl guaiacol in beer is reportedly 200–400 ppb[38]. Here, the Guinness yeasts: 1950, 1959BY, Ikeja and 1981 yeast all produced 4-vinyl guaiacol at concentrations below the flavour threshold limit, whereas, the 1947, 1960 and Park Royal 1960 yeast all produced 4-vinyl guaiacol at a mean concentrations >1000 ppb.

The POF phenotype performs an important environmental fitness function for wild *S. cerevisiae* as it enables the yeast cell to detoxify the phenylacrylic acids present in plant cell walls[39,40]. The genes *PAD1* and *FDC1*, encoding a phenylacrylic acid decarboxylase and a ferulic acid decarboxylase respectively decarboxylate the phenylacrylic acid, ferulic acid, to 4-vinyl guaiacol[40]. Illumina sequence analysis of the Guinness yeasts identified two SNPs in *FDC1* gene, and 9 SNPs in *PAD1* (8 homozygous and 1 heterozygous). Of the 11 identified SNPs, 9 SNPs have been previously identified in other strains of *S. cerevisiae*[40]. The two SNPs identified only in the Guinness yeast are the heterozygous SNP at position 425 in *PAD1* gene and the homozygous SNP at position 790 in *FDC1*. The identified non-synonymous changes do not result in a loss of POF production function.

The median average of the CNV of *PAD1* and *FDC1* within the Guinness yeast was 4. Examination of the data showed that CNV and 4-vinyl guaiacol occurrence in beers were not related for the production for the Guinness yeast studied here (f-test P = 0.86; t-test P = 0.17) POF; yeasts with identical SNPs and CNV within *PAD1* and *FDC1* genes produced different concentrations of 4-vinyl guaiacol under identical experimental conditions.

**Flocculation phenotype of the Guinness yeast.** Yeast flocculation is a reversible, non-sexual aggregation of cells which is of benefit to brewers since it improves the efficiency of sedimentation or separation from beer at the end of fermentation[41]. The timing of flocculation in *S. cerevisiae* is dependent upon expression of the flocculation genes *FLO1, FLO5, FLO8, FLO9, FLO10* and *FLO11* and environmental factors such as calcium, pH, temperature, fermentable sugars and other nutrients[42–47]. For the

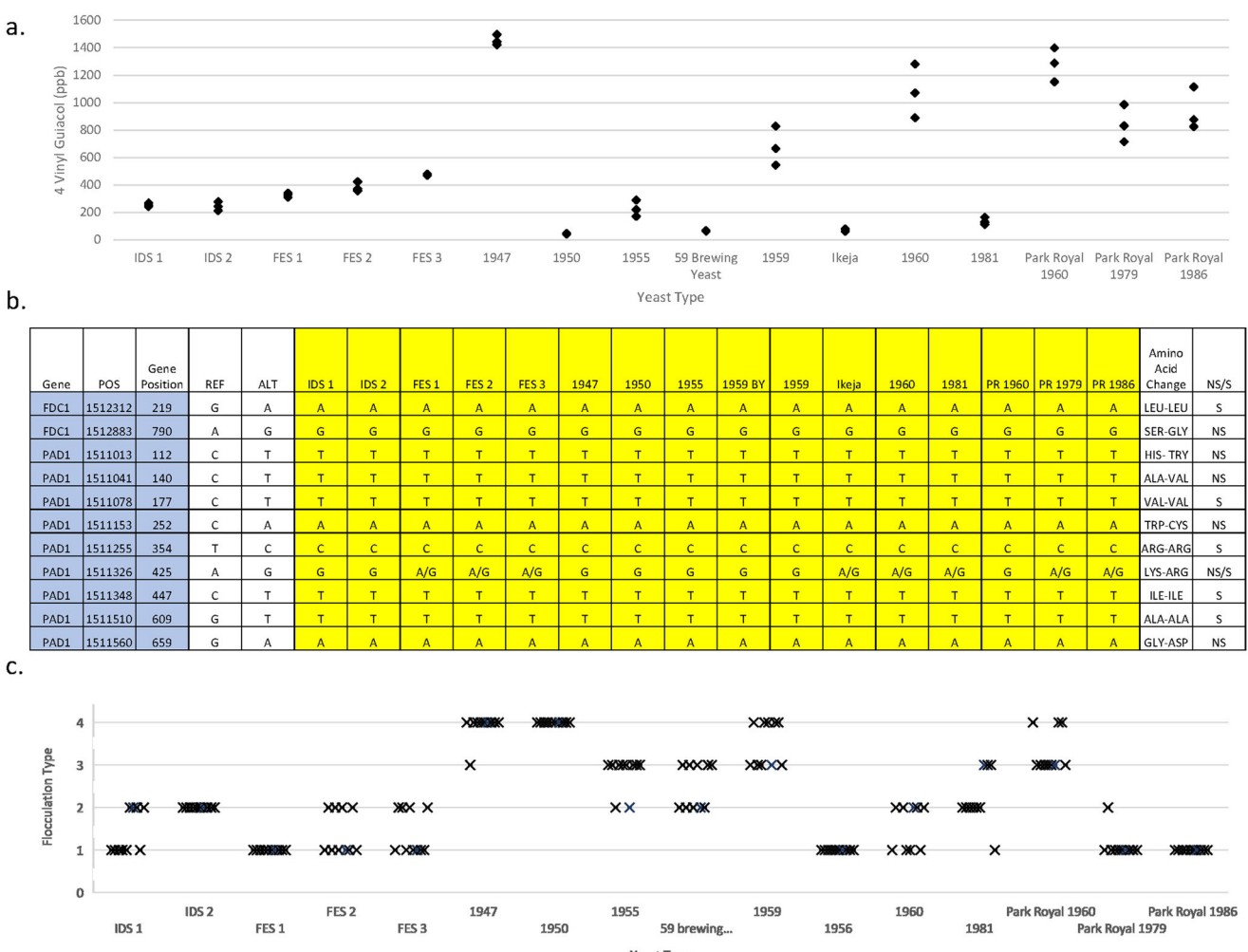

**Fig. 6 Phenolic off-flavour and flocculation phenotype of the Guinness yeast. a** Phenolic Off Flavours (POF) production as determined by the Analytica-EBC Method 2.3.9.5[97] and Gas chromatography mass spectrometry, and **b** Illumina sequencing data of the single nucleotide polymorphism mutations of the POF genes *FDC1* and *PAD1* of the different Guinness yeast. The effects of the SNP mutations result in amino acid substitutions that are non-synonymous (NS) or synonymous (S). Each POF observations presented are *n* = 3 biologically independent experiments. **c** Flocculence characteristics of the different Guinness yeast as determined by the Analytica-EBC Gilliland Method EBC 3.5.3.1[48]. The method class yeast flocculence in terms of non-flocculant (Class 1), slightly flocculant (Class 2), moderately flocculant (Class 3) and highly flocculant (Class 4). Observations presented are *n* = 10 biologically independent experiments.

Guinness yeasts, the degree of flocculation observed was strain-specific (Fig. 6).

Gilliland[48] developed a method for describing the flocculence characteristics of brewing yeast. Four classification types were defined: Class I non-flocculant, Class II slight flocculant, Class III moderately flocculant and Class IV highly flocculant. The production yeast IDS and FES were shown to be Class II and Class I respectively, confirming that the flocculation phenotype of the current Guinness production yeast is the same as those selected in 1959 and 1960 (Fig. 6c).

Nanopore assessment of the IDS1 Guinness production yeast established that three complete *FLO* genes, *FLO9, FLO11* and *FLO8* were present. Scaffolding of the other Guinness yeast using IDS1 as the reference genome confirmed the presence of the three *FLO* genes indicating that *FLO9, FLO11* and *FLO8* are common to the Guinness yeasts. Of the incomplete *FLO* genes, *FLO1, FLO5* and *FLO10*, a truncated version of *FLO1* was identified. A partial read of *FLO5* was identified in the sub-telomeric region of chromosome VIII. The presence of *FLO10* could not be confirmed as the sub-telomeric region of chromosome XI was

absent in the nanopore sequencing data resulting in the loss of *FLO10* and the adjacent genes *VBA5, NFT1* and *GEX2*.

The *FLO* genes identified in the Guinness yeast encode flocculins, *FLO9* and *FLO11*, and the flocculation transcription factor *FLO8*. Flocculation phenotype is affected by the length of the flocculin molecules which projects from the cell wall surface. The longer the flocculin the stronger flocculation competence[49]. The size of the *FLO* encoded flocculins is a consequence of the number of tandem repeats within the ORF[49,50] (Supplementary data 4). The *FLO11* ORF in the Guinness yeasts transcribes a 744 amino acid protein, in comparison that of the reference *S. cerevisiae* S288C genome comprises 1367 amino acid residues[51]. Furthermore, *FLO9* contains two ORF encoding two fragments of the flocculin gene. For the Guinness yeasts FES 3, Ikeja, 1960 pitching yeast, and the 1979 and 1986 Park Royal yeasts a heterozygous SNP mutation at position 12,114 induces a premature stop codon, all the other Guinness yeast retain the consensus SNP resulting in a functioning ORF. The five Guinness yeast that carry the heterozygous SNP in *FLO9* have a flocculation phenotype of <1.5 with three of the yeast observed to be Class I and therefore described as being non-flocculant. The *FLO8*

transcription factor is functional in all of the Guinness yeasts examined here.

## Discussion

Our analysis established that the Guinness yeast form their own subgroup within the previously described Beer 1 brewing clade and that the grouping of the Guinness yeast is separate from other historical Irish Brewing yeast strains. The Beer 1 clade SNPs are of European and Asian origin[9,10] and unlike the other historical Irish Brewing yeast that group within the 'Britain' subpopulations, the data presented in this study indicates the contribution of several lineages to the genetic make-up of the Guinness strains. These different lineages presented in this study establishes that the Guinness yeast are mosaic with an ancestry <80% from a single geographical population.

The analysis presented in Fig. 1 and Supplementary Fig. 2 establishes that the yeast Beer042 shares a recent common ancestor with the Guinness yeast. The Beer042 was deposited in 1979 and at that time was used to brew a lager-style beer in Belgium (personal communication with the owners of Beer042). Beer042 was deposited using the then accepted nomenclature for lager strains *Saccharomyces carlsbergensis* Hansen. Subsequent whole-genome sequencing has confirmed that it was mislabelled and that it is a *S. cerevisiae* yeast. Within the Guinness archive, the last mention of yeast being brought into the brewery is on the 2nd of January 1854. The performance of this yeast was described as poor, consequently it was disposed of, and brewing continued using the then house yeast. There are no subsequent entries of additional yeast in the Guinness archives, other than the Guinness yeast, being used to brew Guinness. As Beer042 was labelled as *Saccharomyces carlsbergensis* Hansen it is likely that it originated from Emil Hansen's Carlsberg group yeast collection. At the end of the 19th century Emil Hansen pioneered the selection and propagation of pure yeast strains for use in brewing[11]. At that time this was a novel concept, and it prompted much interaction between various European brewers[52]. In the case of Guinness, company scientists visited numerous breweries in the UK on multiple occasions[53]. Consequently, a plausible reason for the relationship demonstrated here with Beer042 and the Guinness yeast is that a common ancestor was shared from Dublin to other European brewers. Originally, this common ancestor would have been deposited in Emil Hansen's collection. The resulting observed differences in SNPs between Beer042 and the Guinness yeast are likely due to the consequences of yeast evolution driven by differences in handling practices.

The phylogeny assessment of the Guinness yeast confirmed the expected genealogical relationship between the historical and current production strains, with a division being observed at a pre- and post- '1959' timeline. The 1960 pitching yeast and Park Royal 1960, were collected in 1960 but are yeasts derived from pre-1959 stock. The selection that was undertaken in 1959 used single-cell isolates obtained from the then Guinness Pitching yeast (Guinness Archives). These isolates were tested in the Guinness Research Laboratory with the flocculation phenotype used as the principal differentiating selection criterion. Subsequently the Class II IDS yeast was chosen to brew Guinness Irish Stout, with a second selection from the single-cell isolates undertaken in 1960. This produced a Class I flocculant yeast which was chosen to produce FES on the basis that it was advantageous for yeast to remain suspended in bottle conditioned stout intended for the export market.

The flocculation assessment of the Guinness yeast reported in this study confirmed that the phenotype of the production yeasts FES and IDS have been preserved when compared with their 1959 and 1960 phenotypes. Whilst the flocculation phenotype of the

production yeast has been maintained the present production yeast have diverged from the 1959 Guinness pitching yeast with regard to other characteristics. The reason for this may be related to changes in the methods used for preserving cultures. Prior to 1986 all yeast cultures were maintained on wort agar slopes stored at 4 °C and sub-cultured every 6 months. This method was standard industry practice up until the 1970s and 1980s. Following the work by Labatt's Brewing Company[54] yeast culture storage in liquid nitrogen was introduced and widely adopted.

The production yeast IDS and FES were subjected to an additional reselection procedure which started with individual non-petite yeast colonies from which phenotypes were chosen that were 'prone to spontaneous changes'[55]. The phenotypic characters of interest were flocculation, maltotriose utilisation and head formation (cropping behaviour at the end of fermentation). Some 50 colonies were selected, pooled, and cultured on fresh agar slopes. The rationale was that this should minimise the risk of selection of a potential defect or mutation from a single source which would adversely affect the Guinness yeast. This strategy was adopted since it was concluded that it would mitigate the potential adverse effects of long-term maintenance of yeast cultures on slopes[56]. This process of selecting positive phenotypic traits could over time increase dissimilarities resulting in potential divergence from the original 1959 Guinness yeast. This process is similar to adaptive evolution, the process of positive selection of an advantageous phenotype. Adaptive evolution has been used successfully by others to enhance yeast phenotypes[31,57,58]. The 'adaptive evolution' hypothesis is further supported by the observable differences in the 1981 IDS Guinness yeast compared to the IDS1 and IDS2 production yeast. The last reselection of the Guinness production yeast took place in 1989, 8 years after the 1981 Guinness pitching yeast was deposited in the yeast library. The 1981 IDS yeast is chronologically the closest yeast to the current IDS production yeast, unlike the IDS yeast, the 1981 IDS yeast was not reselected therefore the 1981 yeast is a record of IDS yeast at that time. The observable hierarchal clustering of the Nanopore phylogeny assessment places the 1981 IDS yeast closest to the IDS yeasts, accordingly the observable differences between the IDS yeasts and the 1981 IDS yeast are likely to be a consequence of the Guinness yeast 'adaptive evolution' reselection process.

The loss of meiotic cell division and observed aneuploidy of the Guinness yeast are congruent with yeast domestication[59]. Aneuploidy can confer phenotypic advantages such as enhanced tolerance to ethanol, temperature and oxidative stresses[60–63]. Although assessment of stress resistance was not in the scope of this study the observed variable ploidy together with the phenotypic assessments made via studies of fermentation performance suggest that the Guinness strains have evolved to manage environmental stresses associated with commercial brewing. For example, acquisition of additional copies of chromosome III which correlated with improved ethanol tolerance has been reported[22]. Others, studying industrial processes employing *S. cerevisiae*, including baking and sake brewing, have suggested there is no evidence of amplification of specific chromosomes carrying traits which can be directly attributable to industrial practices[63]. The studies reported here support the latter contention. For the strains examined, no specific chromosome was identified as being responsible for the Guinness yeast phenotype.

In addition to ploidy, gene copy number correlates with gene expression. Aneuploid yeast with multiple gene copy numbers will have increased expression levels compared to a haploid yeast with a single copy gene[22,61]. Increased gene copy numbers does not always result in an increase in the concentrations of the resultant proteins, even though the genes are translated since turnover via proteolysis also occurs[64]. Data presented in this

study established that there are gene CNVs between the different Guinness yeast strains but with reference to the concentrations of important beer flavour yeast-derived metabolites: higher alcohols, diacetyl and phenolic off-flavour (POF), the CNV did not significantly influence the phenotypic outcome.

The observed difference in phenotype between the Guinness yeasts are of interest to brewers as the data presented in this study establishes that yeast that are genotypically similar can be phenotypically diverse. The phenotypic data presented in this study does not correlate with the group of yeast. For example, there are differences between the Park Royal Guinness yeast, present production yeast, pre-1959 and post-1959 Guinness yeast. The difference in POF, esters, higher alcohols and ethanol production is yeast strain-specific. Predicting phenotype based upon genotype is challenging[65–68], even with the use of machine learning predicting phenotype based upon genotype[69] has a poor correlation for *S. cerevisiae* (<22%). The data presented in this study provides further valuable awareness of the relationship of genotype and phenotype and confirms previous studies' conclusions on the difficulty of predicting phenotype from genotype.

Good brewing practice is to replace pitching yeast; the yeast added to wort, after 8–15 re-pitching procedures[70,71]. A major reason for this practice is to avoid genetic drift so that the fermentation outputs remain consistent[72]. The data presented in this study offers another potential explanation; especially for pitching yeast that are not pooled from a pure culture. The differences in fermentation behaviour maybe a consequence of the difference in phenotype. The Guinness production yeast, IDS and FES contain yeast that are phenotypically diverse although they are genotypically similar, consequently, as the yeast is re-pitched there is potential for the concentration of the different yeast to change resulting in a different fermentation/phenotype response. The observations of phenotype and genotype in this study raises important questions for brewers and other groups that use *S. cerevisiae* yeast for industrial processes and highlights the importance of brewers maintaining their production yeast. Further consideration should be given to understanding the role of genotype on phenotype as this will improve *S. cerevisiae* industrial comprehension.

All Guinness yeast strains examined had a POF[+] phenotype. This phenotype is widely found within the wild *S. cerevisiae* population but much less common in industrial strains[4]. The same authors and others have reported that none of the examples of the Britain, Ireland and USA brewing yeasts assessed were POF[+] [4,8]. This suggested that the loss of POF production is a consequence of deliberate selection by the brewer. In contrast, where the POF flavour is an important characteristic; as with German Hefeweizen beers (wheat beers), the POF phenotype has been retained[4,8]. In the case of the Guinness yeast retention of the POF genotype is unusual for domesticated brewing yeast[4,8].

Retention of the POF phenotype in the Guinness yeasts was unlikely to have been a deliberate act; rather at the concentrations found in the beers the effect was benign as it did not create flavour issues. The precursor of 4-vinyl guaiacol, the causative agent of POF, is ferulic acid a component of cell wall polysaccharides of barley, wheat, rice and maize[38]. Free ferulic acid is released during the mashing stage, a process used by brewers to convert grain starches into fermentable sugars during wort production. The free ferulic acid is then converted to 4-vinyl guaiacol by POF[+] yeast during fermentation. The extent of ferulic acid release is influenced by the conditions employed during mashing[73]. A mash temperature stand of 45–50 °C is optimal for releasing ferulic acid[24]. This is a typical starting temperature for Continental European brewers and consequently this suits those beers that have a pronounced POF character, otherwise POF[−] yeast strains must be used[74]. Irish and British brewers prefer

isothermal or infusion mashing using a temperature of 67 °C[74]. This regime does not favour extensive release of ferulic acid[74] and therefore the potential for development of POF in Guinness stouts would not be great. The importance of the presence of POF in Guinness beers is further reduced since an internal expert sensory panel has determined that the flavour threshold concentration for 4-vinyl guaiacol in Guinness stouts is higher than in other beer styles. These factors in concert would reduce pressures to eliminate POF genes in Guinness production yeast.

All Guinness yeasts are POF[+] but the concentration of 4-vinyl guaiacol formed varies with individual yeast strains. The Guinness yeast all share the same SNP mutations and the CNV number does not affect POF production. The presence of stop codon in *FDC1* and *PAD1* result in the loss of POF in negative yeast strains[4,8]. However, Gonçalves et al.[8] observed that in the yeast TUM 507 stop codons present in *PAD1* and *FDC1* did not result in loss of POF phenotype, moreover, in the TUM 380 yeast where there were functioning *PAD1* and *FDC1* genes present, there was a loss of POF production phenotype[8]. Gonçalves[8] concluded that as yet unidentified compounds or enzymes were affecting POF production in TUM 380 and 507 yeasts. Observations made in this study may corroborate Gonçalves's[8] conclusions since individual Guinness strains produce significantly different concentrations of 4-vinyl guaiacol even though all yeast share the same gene content. These observations warrant further investigation.

This study provides evidence of beer style influencing yeast selection. With the exception of Hefeweizen specific yeast, previous studies have demonstrated that yeast subdivide along geographical locations and not beer style[4]; our findings run contradictory to this observation. The data presented in this study establishes that yeast from a similar geographical location, Ireland, are genealogically dissimilar despite documented evidence of the wide sharing of yeast between brewers. All of the non-Guinness Irish brewing yeast were used to brew ales, whilst the Guinness yeast were used to make stout. The phylogeny assessment of the Irish yeast divide on the stout/ale brewing axis. Perhaps the reason for the difference is that yeast used to brew ales are generalist, brewing different types of beers, consequently yeast with universally preferable characteristics such as good flocculation and POF[−] would be selected for by the brewer. For yeast that brew a specific beer style these universal characteristics are not essential subsequently brewers can select a yeast that enhances the features of a particular beer style even if these characteristics are unsuitable for generalist yeast. This hypothesis is supported by the findings of both Gallone et al.[4] and Gonçalves et al.[8] who observed that yeast used to brew beers in the Hefeweizen style are distinct, forming their own subgrouping; interestingly like the Guinness yeast Hefeweizen yeast are also mosaic[4,8]. The effects of raw material, especially the mineral content of water are well known[24]. This has led to the association of geographical locations with certain beer types such as Burton on Trent in the United Kingdom with ales, and delicate lagers associated with Pilsen in the Czech Republic[24]. Unlike brewers vintners use the microflora of the raw material grapes to ferment. Studies have demonstrated that the microflora of geographical regions are associated with certain types of wine[75] this has resulted in the concept of terroir. For wine terroir is well-established but the concept of terroir in beer is still in its infancy despite the recent studies on the terroir of hops[76,77]. Our findings provide another possible avenue for brewers to further explore the concept of beer associated terroir.

In conclusion the analysis presented in this study establishes that the Guinness yeast are not only significantly different from other historical Irish Brewing yeast but they form a sub- group within the brewing yeast clade. The genealogy of the different

Guinness yeast is confirmed by our analysis and supports the Guinness archive historical records that the Guinness yeast used today is related to the first deposited Guinness yeast; the 1903 Watling Laboratory Guinness yeast.

## Methods

**Yeast strain selection and maintenance**. A total of 19 Irish brewing yeast strains including 13 from the Guinness collection were selected for assessment (Table 1). The Guinness strains included two current production strains and 11 other historical strains selected as they were the principal brewing strains used to produce Guinness at that time. Cultures were stored in cryo vials (Fisher) in liquid nitrogen at −196 °C using 50% glycerol (Sigma–Aldrich) as a cryo-preservative. Cultures were recovered and inoculated into 25 ml tubes containing 10 ml of YPD (10 g l$^{-1}$ yeast extract, 20 g l$^{-1}$ peptone, 20 g l$^{-1}$ glucose) (Oxoid) and incubated at 25 °C in an orbital shaker (Stuart Scientific) at 120 rpm for 24 h. Serial dilutions of 100 µl of cultures were spread plated onto Wallerstein Nutrient Agar (Oxoid) and incubated at 25 °C for 12 days in accordance with the EBC Yeast Giant Colony method 3.2.1.1[78]. At the end of the incubation, single yeast colonies were selected.

**DNA extraction and interdelta yeast typing**. Three giant colonies of each culture were selected and transferred to microfuge tubes containing 700 µl of molecular grade water (Fisher). Yeast cells were recovered by centrifugation and DNA was extracted in accordance with the manufacture's guidelines using a PureLink Microbiome DNA Purification Kit (Invitrogen). Individual strains were identified using the interdelta (ITS) Polymerase chain reaction PCR method[14,79]; primers δ2 (5′-GTGGATTTT-TATTCCAAC-3′) and δ12 (5′-TCAACAATGGAATCCCAAC-3′), using a BioRad T100 Thermocycler and Invitrogen's Platinum Hot start PCR Master Mix. PCR products were analysed on an Agilent 2100 Bioanalyzer using the Agilent DNA 7500 chip. The resulting bands were analysed using Minitab 19 Statistical software (2019) hierarchical clustering function with dendrograms produced using Euclidean distance function.

**Illumina whole-genome sequencing and de novo assembly**. The ITS results were used to select strains for whole-genome sequencing with typical ITS banding used as the selection criterion for the historical Guinness and Irish brewing yeast. In the case of the FES and IDS production yeast all yeasts that were determined to be unique were selected. In total, 16 Guinness strains and 6 other historical Irish brewing yeast were subjected to whole-genome sequencing performed by Elda Biotech (Kildare, Ireland). Yeast samples were sub-cultured onto Wallerstein Nutrient Agar (Oxoid) and single colonies were picked for DNA extraction using the Thermo Scientific Yeast DNA extraction kit (Thermo Scientific). Extracted DNA was analysed using a Qubit (Thermo Fisher Scientific) to determine dsDNA content. Aliquots of 1 ng of DNA was used as input for library preparation using the Illumina Nextera XT DNA library prep protocol with no deviations. Stock libraries of 1–4 nM were generated and samples were pooled for sequencing and denatured according to the manufacturer's instructions for loading on the Illumina MiSeq (12 pM) sequencer. Samples were sequenced using the Illumina MiSeq machine reading with a minimum depth of 30× coverage using $2 \times 250$ bp paired reads. All samples were quality checked for low-quality sequence bases and the presence of adaptor contamination using Trimgalore (Version 0.6.1). All identified adaptors were cleaved from both the forward and reverse sequencing reads and those with runs of low-quality bases were trimmed using a Phred scale cutoff of 10. All samples were

aligned to the reference genome *S. cerevisiae* S288C (http://downloads.yeastgenome.org/sequence/S288C_reference/genome_releases/S288C_reference_genome_R64-1-1_20110203.tgz) using BWA (Version 0.7.17)[80]. Alignments were sorted and duplicate reads were identified and marked for exclusion from downstream analysis using Samtools (Version 1.10)[81]. Alignment metrics for each sample were collated using Qualimap (Version 2.2.1)[82]. All samples were de novo assembled using Spades (Version 3.14). For each sample all contigs shorter than 500 base pairs in length were discarded. A reference guided scaffold of each assembled sample genome against the *Saccharomyces cerevisiae* S288C genome sequence was generated using Ragtag (v. 1.0.2)[83] Artificial padding of "N" characters was placed between the reference scaffolded contigs. All bioinformatic software used in this study is specified in Supplementary Table 1.

**Nanopore MINION sequencing**. Two nanopore Minion runs of Irish Draught Stout yeast number 1 (IDS1) were carried out by ELDA Biotech (Kildare, Ireland). DNA from an IDS1 colony grown on Wallerstein Nutrient Agar (Oxoid) was extract using the Thermo Scientific Yeast DNA extraction kit (Thermo Scientific). Extracted DNA was processed using the 1D Genomic DNA by ligation (SQK-LSK108) protocol from Oxford Nanopore Technologies. DNA was fragmented using a Covaris g-TUBE (Covaris) with DNA repair performed using End Prep (New England Biolabs). Library clean up and adaptor ligation were performed with AMPure XP beads (Beckman Coulter), and extracted DNA measured using a Qubit (Thermo Fisher Scientific). 3.6fmol of DNA library was loaded onto the flongle for sequencing (Oxford Nanopore Technologies). Sequencing was conducted according to the Nanopore Minion manufacturer's instructions (Oxford Nanopore Technologies). All Minion run Fast5 files were converted to FASTQ format using Guppy (3.6). NanoFilt was used to remove low-quality reads (Q10) and reads shorter than 1000 bases, with Porechop v6[84] used to find and remove adaptors located at the start, end or internal reads. Contaminant (non-fungal) reads were identified using both Kraken2 and BLAST searches against the NCBI non-redundant database. Identified contaminant reads were removed using Seqtk (v1.3). The first Nanopore Minion run resulted in 421,040 usable reads and 64,105 in the second. Reads from both runs were combined for a unified assembly using Flye (v. 2.8)[21]. A corrected consensus sequence for the Flye assembly was generated using Medaka (v 1.0.3). Racon was used as a polishing tool for the Medaka consensus sequence using the previously generated IDS1 Illumina data and an assembly evaluation using Quast 5.10.0 was carried out using this nanopore assembled genome and the previously assembled Illumina only IDS1 genome. Gene content completeness of the assembled genome was estimated using Busco (v3)[85] with the assembled nanopore genome scaffolded against the *S. cerevisiae* S288C reference genome using Ragtag (v. 1.0.2)[83]. Finally, the scaffolded assembly was annotated using Funannotate (v 1.74)[86].

**Determination of the Guinness yeast phylogeny**. Sequencing data for 154 *S. cerevisiae* samples[4] were retrieved from NCBI (BioProject PRJNA323691) and combined with the 22 *S. cerevisiae* sequenced for this investigation. All retrieved samples were quality checked for low-quality sequence bases and the presence of adaptor contamination using Trimgalore (Version 0.6.1). All identified adaptors were cleaved from both the forward and reverse sequencing reads and reads with runs of low-quality bases trimmed using a Phred scale cutoff of 10. All samples were aligned to *S. cerevisiae* S288C reference genome using BWA mem (version 0.7.17)[80]. Alignments were sorted and duplicate reads

were identified and marked for exclusion from downstream analysis using Picard (Version 2.18.23). Alignment metrics for each sample were collated using Qualimap (Version 2.2.1)[82]. Misalignment of reads in original BWA alignments were corrected using GATK (Version 4.1.4-1)[87] with the base score recalibration carried out on the corrected alignments. SNP and Indel discovery and genotyping was performed across all 176 samples simultaneously with GATK used to filter sites based on the following metrics: quality score >30, mapping scores >40, read position rank sum <8. All individual genotypes with less than 10× coverage were set to uncalled. Annotation and effect prediction for each variant was estimated using SnpEff (Version 4.3)[88].

Orthologous genes across all assembled genomes were inferred using Orthofinder (Version 2.3.3)[89]. Sequences from orthologous genes were concatenated and aligned using MUSCLE (Version 3.8.31). A phylogenetic analysis of the concatenated alignment of data from all orthologous genes was carried out using the maximum-likelihood approach implemented in RAxML (Version 8.2.4)[15] based on the GTRGAMMA model of sequence evolution and a rapid bootstrap analysis for 1000 bootstrap replicates. The tree which was rooted using the outgroup species *S. paradoxus* was visualised and annotated using the ggtree[16] package in R.

FastSTRUCTURE (Version 1.0)[17] was used to quantify the number of populations and the degree of admixture in the genomes examined in this study. Owing to the high degree of sequence similarity between the Guinness samples a single representative sample (IDS1) was used in this analysis consequently 161 genomes admixture were assessed. The full set of biallelic segregating sites identified across all samples was filtered based on a minor allele frequency (MAF) < 0.05 and SNPs in linkage-disequilibrium, using PLINK (v1.09)[90]. FastSTRUCTURE[17] was run on a filtered set of SNPs, varying the number of ancestral populations (K) between 1 and 10 using the simple prior implemented in fastSTRUCTURE[17] with $K = 8$ found to be optimal. The admixture of IDS1 was determined from the sequence data of the 154 *S. cerevisiae* samples[4] using Alpaca (v1)[18] and a kmer length of 21 over 5000 base pair sliding windows.

**Copy number variation**. Analysis of the heterozygous biallelic SNPs for each Guinness yeast established variable copy number across the chromosomes and subsequently CNVs was normalised against an appropriate background copy number for each strain. In addition, CNVs was estimated in 250 base pair non-overlapping windows across the entire ~12 million bases of the *S. cerevisiae* genome using Control-FREEC (Version 5.7)[91]. Plots depicting CNVs for the Guinness yeasts were generated in R using publicly available code[92].

**Sporulation**. The sporulation potential of the different Guinness yeast strains was assessed using the ASBC Yeast 7 sporulation method[93]. A total of 1000 cells per sample were examined using a Nikon Eclipse C*i* microscope 100× magnification. Ascospores stained green to blue green while vegetative yeast cells-stained pink to red. independent triplicate analyses were performed for each strain. The incidence of sporulation was expressed as a percentage.

**Assessment of fermentation properties**. Fermentation ability was assessed using 180 ml mini fermenters (Fisher) containing 120 ml of 12ºP wort. Cultures were recovered from liquid nitrogen and sufficient yeast for the experiments generated by successive serial aerobic incubations in 10 ml YPD, 90 ml 12ºP wort and 900 ml 12ºP wort. A single batch of all-malt hopped

wort was used for all experiments to eliminate batch to batch variation. Wort was produced in the Guinness Pilot plant and stored at −20 °C in 5 l aliquots. Prior to use it was thawed and sterilised by autoclaving.

Yeast cells were recovered by centrifugation and washed three times by successive suspension in distilled water and re-centrifugation. Viability and yeast cell concentration of each culture was determined using the EBC methods, EBC 3.1.1.1 Haemocytometer[94] and EBC 3.2.1.1 Methylene Blue[95]. Triplicate fermentations were inoculated with $1 \times 10^7$ viable yeast cells per ml into 180 ml mini fermenters containing 120 ml of air-saturated 12ºP wort. Fermentations were incubated at 25 °C and stirred continuously using a stirrer plate (mix 15 eco plate Camlab) set at 250 rpm. Mini fermenters were sealed with a butyl rubber plug secured with an aluminium cap (Fisher) and fitted with a Bunsen valve to allow $CO_2$ to be released. Fermentation progression was measured by periodically monitoring weight loss. The endpoint was established when three successive identical readings were recorded.

**Analysis of fermentation metabolites**. Concentrations of selected yeast-derived flavour compounds were measured using a gas chromatographic procedure using a modified version of the EBC Vicinal Diketone method Analytica-EBC Method 9.24.2[96]. End-fermentation samples (30 ml) previously clarified by centrifugation were transferred to McCartney bottles which after sealing were heated at 65 °C for 30 min to convert precursor α-acetolactate into free diacetyl. Diacetyl (2,3 butanedione) concentration was determined using an ECD detector; with esters and higher alcohol concentrations determined using an FID detector. Peak areas for the metabolites were normalised using appropriate internal standards.

**Analysis for phenolic off-flavour (4-vinyl guaiacol, 4-VG)**. The ability of yeast to produce 4-vinyl guaiacol was determined according to Analytica-EBC Method 2.3.9.5[97] phenolic off-flavour method using gas chromatography mass spectrometry. Washed yeast samples were inoculated at a concentration of $1 \times 10^6$ viable cells ml$^{-1}$ into 25 ml tubes containing 10 mls of YPD medium supplemented with 0.1 ml of ferulic acid (hydro-xycinnamic acid) solution. Triplicate incubations were performed for each yeast strain. After incubation at 25 °C for 48 h. 5 ml was transferred to an autosampler vial (Fisher) containing 2 μl of the internal standard containing: 4-vinyl guaiacol (Sigma–Aldrich). Analyses were carried out using an Agilent 6890/7890 GC systems fitted with a Zebron ZB-Wax 60.0 m × 250.00 μm × 0.25μm column. The initial oven temperature was 60 °C. After 10 min this was increased to 220 °C at a rate of 10 °C min$^{-1}$ then held for 2 min. 4-vinyl guaiacol concentration was determined using an ECB detector; temperature 150 °C, make-up flow rate of 60 ml min$^{-1}$ (Helium gas). Peak area for 4-vinyl guaiacol were normalised against the internal 4-vinyl guaiacol standard.

**Alcohol concentration**. Ethanol concentration was determined using near infrared spectroscopy using an Anton Paar Alcolyser in accordance with the manufacture's guidelines.

**Sugar concentration**. Samples were analysed tested using an Agilent 1260 Infinity II system with a refractive index detector (Infinity II 1260 WR RID) and a Zorbax Carbohydrate column (4.6 × 250 mm, 5 μm, P/N: 840300-908). The other acquisition conditions were as follows: mobile phase was a 70/30 mix of acetonitrile and water; sample injection volume was 50 μL; flow rate was isocratic and set at 1.5 ml min$^{-1}$. The column oven was kept at a constant 35 °C. No internal standard was used. However, samples were bracketed either side with freshly made known standard.

**Flocculation**. Flocculation was assessed using EBC Gilliland method EBC 3.5.31[48]. The EBC method uses visual inspection of flocculation behaviour categorising the yeast using prescribed classifications: Class 1 non-flocculant, Class 2 slightly flocculant, Class 3 moderately flocculant and Class 4 highly flocculant. An addendum to the EBC method was the addition of four control yeasts representing the different classifications.

**1HL fermentations**. In order to prepare sufficient beer for taste testing 1 hL fermentations using selected yeast strains were carried out using the Guinness pilot scale plant. Standard Guinness wort was taken from the St James' Gate Brewery and diluted to 12ºP with deaerated water and autoclaved prior to use. Yeast strains were retrieved from long-term liquid nitrogen and propagated by successive serial aerobic incubations in 10 ml YPD, 90 ml 12ºP wort and 900 ml 12ºP wort. To ensure that sufficient yeast was available the terminal cultures were prepared in a Carlsberg flask (GEA) containing 15 litres of sterile 12ºP Guinness wort and incubated at 25 °C for 48 h with continuous oxygenation. This generated sufficient yeast to inoculate 80 L of wort at an initial count of $1 \times 10^7$ viable yeast cells ml$^{-1}$. Fermentations were attemperated at 22 °C. After the desired final gravity was achieved the beer was held at 25 °C for 24 h to allow removal of diacetyl. A 20 L sample of beer was then removed and transferred to a sterile keg. After storage at 4 °C for 48 h the beer was clarified by passage through a sheet filter then bottled and pasteurised (25 PU). Triplicate samples of the beers were analysed for alcohol, fermentation metabolites, and POF production using an Anton Paar Alcolyser, gas chromatographic procedure using a modified version of the EBC Vicinal Diketone method Analytica-EBC Method 9.24.2[96] and Analytica-EBC Method 2.3.9.5[97], phenolic off-flavour method using gas chromatography mass spectrometry. Organoleptic properties was assessed by a trained beer sensory panel consisting of 18 members. Using Quantitative Descriptive methodology[98] and a list of predefined Guinness Stout sensory attributes all three samples were tested in duplicate by the panel in a single tasting session. Individual sensory attributes were rated on a linear scale (0–10)[99], with a subset of these attributes identified to explain differences and similarities across samples. The samples were randomised and presented to the taste panel labelled with a three-digit code. Sensory scores were analysed using a 2-way ANOVA including sensory assessor's as a random factor[99].

**Statistics and reproducibility**. Statistical analyses were preformed using Minitab 19 Statistical software (2019) and Xlstat 20 Excel statistical package (2020). Mini-fermentations were performed using three independent biological replicates and statistical significance of ethanol production, fermentation metabolites and POF determined using one way ANOVAs. The organoleptic properties of the 1Hl fermentation Guinness brews were determined by Quantitative Descriptive methodology[98] using a trained taste panel of 18 independent members, and the resulting data analysed using 2-way ANOVA. T-test were performed on iso-butanol production of IDS1 and IDS2. The effects of CNV on POF production was established using $f$ and $t$-tests.

**Reporting summary**. Further information on research design is available in the Nature Portfolio Reporting Summary linked to this article.

## Data availability
Illumina and Nanopore (basecalled, demultiplexed) reads for all sequenced samples in this manuscript are deposited in the European Nucleotide Archive (ENA) under the project accession PRJEB62101. All experimental data underlying figures are presented in Supplementary Data 5.

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

## Acknowledgements

The authors would like to thank the Guinness Brand Team for sponsoring the work. The historic records presented in this study were curated with the help of former Guinness Group Microbiologists Dr Edward Bourke. The authors would like to thank Dr Kieran Joyce and his team for their analytical support and Marcus Bengelstorff and Patrick Kerr for their brewing expertise. D.K. and P.C. would like to thank Dr Brigida Gallone, Dr Jan Steensels, Prof Kevin Verstrepen et al. for their 2016 investigation. The St James's Gate Yeast Library has been maintained since 1903 by numerous Guinness' Microbiologists however the authors would like to give special thanks to June Hurley, Dr Barbara Cantwell, Dr Daniel Donnelly, Angela Larkin, Dr Vidya Dixit and Noel Early.

## Author contributions

Illumina and Nanopore sequencing were performed by E.K. with bioinformatic analysis done by P.C. Phenotypic analysis was conducted by D.K. and J.K. Historical archive analysis was provided by E.C. D.K designed the study with bioinformatic experimental design made with P.C. and E.K. D.K. wrote the manuscript with editorial input from E.K., P.C., C.B. and S.S.

## Competing interests

Daniel Kerrruish, Jessica Kearns, Eibhlin Colgan, and Sandra Stelma are employees of Diageo Ireland, the owners of Guinness. Chris Boulton is employed as a consultant by Diageo Ireland. All other authors declare no competing interests.
