## [Peer Review File · Communications Biology]

Reviewers' comments:

Reviewer #1 (Remarks to the Author):

This manuscript reports the genomic and phenotypic/metabolomic analyses of a series of beer yeast, specifically stout yeasts, from the Guinness beer yeast collection. These strains were whole-genome sequenced (all with short-reads [Illumina] and one with long-reads [Nanopore]), and the data used to provide insights into genealogy, phylogeny, populations structure, CNVs and SNPs of relevant genes in beer production. Phenotypic analyses via sporulation, flocculation and fermentation performance along with subsequent metabolite/flavour compound analyses provided insights into the yeasts used for the production of these highly popular beers. The manuscript report these Guinness yeasts as a unique sub-clade within the well-known Beer 1 beer yeast clade. It also reports specific, but unique domestication traits for these yeasts. Interestingly, the range of yeasts from the historical culture collection and associated records allowed the authors to track genealogy; this resource is invaluable and the authors provided some interesting insights in their analyses. To my knowledge, this is the first report of its kind on these specific beer yeasts. In my opinion, this report will attract lots of interest in the fields of beer yeast and beer yeast genomics.

The manuscript provides an interesting story with many historical insights that help in the understanding of the yeasts' development/domestication. The manuscript is interesting, the data compelling and conclusions for the most part solid. Below I provide a few comments and suggestions for the authors to address/consider.

Specific comments:

1. Please confirm the names of the software used in Lines 213 ("10FastSTRUCTURE") and 217 ("11PLINK") as accurate.
2. The names of the yeast strains listed in Table 1 does not align with those used in Supplemental Data 1. Please align the latter with the former to allow easy and accurate tracking by the reader.
3. Line 381-382: "Varying the number of ancestral 382 populations (K) between 1 and 10, K = 8 was found to be optimal (Figure 1)." I could not find the population analyses (K) in Figure 1. PCA, yes, but not evidence of K = 8. Please add this analysis as Fig. 1C or Supplemental Information.
4. Lines 441-444. As all the other metabolite/flavour compound data are supplied in a single bar graph, it would be more useful and easy to compare the data if all the ethanol data can be reported in a single bar graph. Also, while the provided statistical analyses proved confirms the ethanol production is statistically significant, it would be more useful to provide statistical data that comment on strain-to-strain variations. I believe such analyses would be beneficial to the manuscript and might provide/support further insights. The same comment applies to the metabolites reported in Figures 4B-D.
5. Line 456: The authors spend quite some time discussing maltotriose metabolism. Did the authors determine the maltotriose consumption profiles or at least the end-point values to provide the relevant phenotypic link?
6. Line 457: multigene loci?
7. Line 489: According to Fig 4B, more than just the three stated strains surpassed 10 ppm for ethyl acetate. Please clarify.
8. Line 526-530. The authors should reference their data as support; no reference currently exists. Furthermore, Figure 4D is not labelled as 2,3 butanedione (diacetyl), and in the legend 2,3 butanedione (diacetyl) is mislabeled as "C."
9. Lines 533-535: What are these SNPs? Show the data. Are they projected to be homozygous or heterozygous? This information was supplied for the PAD1/FDC1 genes in this manuscript, and should also be supplied for all genes where SNPs are reported.
10. Line 580-583: "The Guinness yeast," FES3 seems to be above 400 ppb and the remaining production yeasts seem to be within the fairly broad 200-400 ppb threshold range. Did the authors only consider the upper limit as the threshold? Similarly, does 1955 also fall below the threshold?

Please clarify.

11. Lines 589-593: The authors should reference their supporting data.

12. Lines 635-640: Show the data.

13. Lines 712-714: Do the data generated and reported here eliminate this possibility? What about BAT2 and ILV2 and its respective CNV/metabolite correlations described in the results? While it might not provide conclusive supporting data, it cannot be dismissed either.

14. Line 722: "higher alcohols"; This statement is not entirely consistent with the description of the data in the results - see lines 508-512. This description does not eliminate the potential of CNVs in higher alcohol production. And "diacetyl"; This statement contradicts the described results - see lines 535-538.

Overall a well-written, compelling manuscript.

Reviewer #2 (Remarks to the Author):

This is a thorough genomic and phenotypic analysis of yeasts used historically and currently in the Guinness brewery in comparison to other Irish brewing yeasts and yeasts from around the world. The main conclusion that the Guinness yeasts from a unique clade within the 'beer 1' clade of yeasts is well supported.

Given the close genetic relationship of the Guinness yeast strains, I find it surprising how diverse they are phenotypically. CNV does not appear to be correlated but some further general description of potential correlates of sequence variation would be helpful. What does this say about phenotypic stability - an important aspect to any industry producing a consistent product?

In terms of the re-pitching and selection regimes being similar to ALE experiments it would be useful to parse the SNPs determined in the phylogeny generated - particularly the pre and post-1959 subgroups as this would help understand how genetic variation changed over time and procedures. No real discussion of loss of heterozygosity as one process of divergence between strain lineages is had which could be part of the evolution of these strains.

A highly detailed analysis that will be of interest to brewing yeast researchers with some interesting highlights for the more general yeast researcher.

Reviewer #3 (Remarks to the Author):

As a fan of stout, I have the pleasure of reviewing Kerruish et al titled "The Origins of the Guinness Stout Yeast".

The authors sequenced a total of 22 (13 Guinness) strains, conducted phylogenomics and phenotyping analyses. The results of manuscript is heavily biased towards the phenotyping but the histories of the strains in my opinion deserved far more attention but unfortunately was somewhat descriptive. I will focus my review on this particular section.

Major comment: Results on phylogeny and population structure were way too descriptive

- L382: Varying the number of ancestral populations (K) between 1 and 10, K = 8 was found to be optimal (Figure 1);

There's no information on K=8 being listed on Figure 1. Figure 1a is phylogeny and 1b is a PCA figure.

- A supplementary figure should be provided on choosing K=8 (by showing delta of different K).

- As the authors pointed out it "was an unexpected observation" that "The non-Guinness Irish brewing strains were located in the 'Britain' subpopulations, whereas, the Guinness yeast occupy their own subgroup outside the USA and 'Britain' subpopulations."

The authors need to show the the Distruct plot to visualise the expected admixture proportions. (<https://rajanil.github.io/fastStructure/>). It is difficult to pinpoint out the exact scenario from text L385-390

- L388 10 yeasts were designated as being mosaic. Are these the Guinness strains? Please clarify

- L496 How is this "20,000" SNPs chosen?

It is a real cliffhanger simply saying "Consequently It can be concluded that the Guinness strains are Mosaic"

There are so much potential with this data, and I suggest authors to perform two additional analyses:

1. Confirm source of "mosaic" / introgression by assembly and identifying the origin of phased subgenomes

(like <https://www.sciencedirect.com/science/article/pii/S0960982222001300>). Or alternatively, simply check

where the genomes are originated from each lineage using TreeMix

(like Fig 3 <https://genome.cshlp.org/content/32/5/864>)

2. Calibrate the phylogeny of Figure 2 to see pinpoint historical events of Guinness strains. See Fig 6 of <https://www.nature.com/articles/s41559-019-0997-9> and Fig 3 of <https://www.nature.com/articles/s41586-020-2889-1>.

And relate the time to source in Table 1. This will greatly improve the scope of discussion at L656-668, boost the impact and scientific interest of this manuscript.

Reviewer #4 (Remarks to the Author):

This paper facilitates the placement of Irish beer yeasts in the phylogeny of industrial yeasts, probably for the first time. The authors show that historical Irish beer yeasts (Smithwicks, Macardles, Perry etc.) and Guinness yeasts belong to the "Beer 1" clade, and more particularly to the "British ales" lineage, which contains isolates from the UK, the US and Ireland, as well as some likely mosaics. This is not particularly surprising. However, the authors also suggest that whereas most of the historical Irish brewing strains are closely related to British strains, the Guinness isolates are "mosaics". Unfortunately, "mosaic" is not well defined, and the supporting data (Structure analysis) is not provided. It is therefore not clear which lineages contributed to the mosaic pattern. There are several other interesting observations. For example, unlike most other "British ale" yeasts, some Guinness isolates have acquired loss of function mutations in a maltotriose transporter, and all Guinness isolates have retained functional FDC1 and PAD1 genes, associated with off flavor (4-vinyl guaiacol) production. I cannot comment on the fermentation assays, which are outside my area of expertise.

The analysis is potentially interesting, but some data is missing, and the mosaic status of the Guinness strains should be explored in more detail. An improved manuscript will be of interest to the general community. The data will also be very useful for the growing field studying the origins and applications of industrial yeasts.

Major points

1. In the introduction, refer to the analysis showing that most beer yeasts result from an admixture of European and Asian wine populations (J.C Fay et al, PLoS Biol 17:e3000147 and Saada et al, Curr Biol. 32:1350).
2. Please improve Fig. 1A, using larger colored dots to make it possible to easily identify the geographical origin of the relevant isolates in Beer 1. Label the mosaics more clearly, and do not include "mosaic" in the same scheme as geographical origin. Can some support values be shown, at least for the Guinness grouping and related lineages? Label the point that divides "British/US/Irish" from "German/Belgian", and then within "British" label the point that divides "British/Irish" from "US/Guinness" (see point 3). Include the Structure analysis in Fig. 1 (see point 5).
3. Gallone et al (2016) and other studies and the current manuscript analysis show that the "Beer 1" group can be divided into two main lineages, the "German/Belgian" lineage, and the "British ales" lineage (British/Irish/US beer isolates), plus some mosaics. The exact names ("German/Belgian" or "German" and "British" or "British/Irish") vary in different publications, but one should be chosen and used in the submitted manuscript. The current analysis shows that all the Irish yeasts (including the Guinness strains) belong to the "British ales" lineage (British/Irish/US beer isolates). Based on Fig. 1A, the "British ales" lineage can be further subdivided into two groups – one contains British isolates plus Irish isolates except for the Guinness isolates, and the second contains US isolates, Guinness isolates, and some unassigned isolates. Both groups contain some mosaic isolates. Without showing support values for the phylogeny or the PCA analysis it is difficult to know how well supported these divisions (and further subdivisions) are. The abstract and the text needs to be clearer about this point.
4. I don't follow the conclusion "The brewing strains of companies Perry, Cherry and Smithwicks aligned completely to the Britain subpopulation; whereas, the Great Northern, Macardles 1966, and Macardles 1993 yeast aligned with the 'Britain' group but also the US and Belgium /Germany subpopulations". Why do you conclude that the Great Northern and Macardles align with the US or the "Belgium/German" subpopulations? This seems to be a misreading of the tree. The Guinness isolates are closer to the US isolates than Macardles from Fig. 1A and 1C, and they are all equidistant from the "Belgium/German" subpopulation. Is there evidence (e.g. from the not shown Structure analysis, see point 5) that Great Northern and Macardles isolates are mosaics? If so, please describe in more detail.
5. The conclusions that the Guinness yeasts are mosaics is hard to follow, partly because the Structure plot is not shown. Please show the results of the Structure analysis with K=8, which might show that the Guinness yeasts are mosaics/hybrids between a British/Irish parent and some other lineage (Belgian/German? Other lineage?). The PCA analysis does not allow the conclusion that the Guinness strains are mosaic (line 389-390). You should also clarify what you mean by "mosaic", particularly in the context that most beer yeasts are admixed anyway (see Fay et al, point 1). Presumably the Guinness strains (and possibly BE42/58, see next point) descended from a more recent admixture.
6. State clearly that the phylogeny in Fig.1 suggests that the closest relatives of the Guinness yeasts are the mosaics/hybrids BE42 (Belgian lager yeast) and BE58 (English beer) (from Gallone et al 2016). The origin of the Guinness yeasts needs to be discussed in more depth. One interpretation is that a single hybridization occurred in the ancestor of BE42/BE58/Guinness isolates, and each lineage then diverged. It is also possible that there were multiple hybridizations. The Structure analysis should help determine if the same parent populations gave rise to all 3 lineages. Phylogenetic analysis of mosaics/hybrids can be difficult.
7. The discussion of the maltose/maltotriose gene loci is confusing. MAL loci are associated with utilization of maltose (for maltotriose, see next point). *S. cerevisiae* genomes differ in the number of MAL loci that they contain, and Beer1 isolates usually have many copies. It should be possible to determine the number and location of the loci in the Guinness IDS1 isolate from the MinION assembly. Please include this in a diagram/figure, which will greatly clarify the description. You can then determine if any of the other Guinness isolates have more/fewer loci by mapping the Illumina reads on the MinION assembly. For example, it is very unlikely that MAL63 has moved to Chr XVI. It is more likely that there is an ancestral maltose transporter on Chr XVI (like there is the reference genome) and that MAL63 has been lost.

8. Most MAL loci encode maltose transporters. However, the MAL11 transporter (also called AGT1) at MAL1 is different. This also transports maltotriose. A frameshift in MAL11 would prevent transport of maltotriose, whereas maltose would still be transported if other MAL transporter genes are intact (line 463). Frameshifts in MAL11/AGT1 are indeed unusual in the "British ales" lineage. Are the frameshifts in the 4 Guinness strains homozygous? You should determine which of the Guinness strains can utilize maltotriose and which can utilize maltose, and see if this correlates with the frameshift in MAL11.
9. The discussion of off flavors (4-vinyl guaiacol) is interesting. Most Beer 1 yeasts (and all "British ales" yeasts) have frameshifts in FDC1 and PAD1 required to make 4VG (Gallone et al 2016, Goncalves et al 2016, Saada et al 2022), as noted in the discussion. One question is therefore where did the intact genes in the Guinness isolates come from? Were they retained from the ancestor of "British ales" yeasts, and acquired frameshifts in every lineage except for the Guinness isolates, and the wheat beers? My interpretation of supplementary material in Gallone et al is that in BE42 and BE58 (closest relatives of the Guinness yeasts) FDC1 and PAD1 do contain loss-of-function mutations, though in one isolate there are heterozygous mutations. This would suggest that FDC1/PAD1 may have been fully functional in the immediate ancestor of BE42/BE58/Guinness. The functional copies could have originated from a Belgian/German parent, where some isolates have intact FDC1 and PAD1 genes (Saada et al. 2022). This is probably worth discussing. Can you indicate the position of the wheat beer isolates on your phylogeny?
10. I can't find a reference to availability of the raw data, though I may have missed it in one of the online documents. The raw data must be available, and the MinION assembly and the annotation must also be provided (if they were available I could look at the MAL loci for example). The custom script (line 222) should also be provided.

Minor -easily addressed

11. The genome data from Smithwicks, Macardles etc. is underused and little background is provided. Could more information about Perry, Cherry, Smithwicks, McCardles etc. breweries be provided? Do these isolates have mutations in MAL1, FDC1, PAD1 etc? Did one of these lineages contribute to the mosaic structure of the Guinness strains?
12. The methods for how the phylogeny was constructed are not provided. A phylogeny cannot be constructed using SnpEff (line 364). The legend of Fig. 1 refers to Maximum Likelihood, but more information is required.
13. One note on using Flye for assembly – this can introduce errors, especially with polyploid eukaryotic genomes. Canu (PMID: 28298431) is often a better choice. You may even have enough raw reads to generate a phased assembly, even for parts of the genome. This would help to determine the origins of the parents of the Guinness strains. (This is a suggestion, not a requirement).
14. Please clarify the number of strains used, which varies from 19 to 22, depending on whether "IDS1" and "IDS2" etc are assumed to be the same or not (e.g. line 116). Make it clear in Table 1 that "Park Royal" are Guinness isolates.
15. Aneuploidy (Figure 3): there is a difference in the copy number of Chr X in IDS1 and IDS2, which should be commented. You may wish to check Chr XV in FES2. It is surprising that part of this is colored black.
16. CNVs: did you account for aneuploidy when calculating copy number? BAT2 is on Chr X, and there are extra copies of Chr X in PR1960. The section on CNV and gene content (from around line 504) is somewhat long and speculative.
17. I can't follow the description of 1981 IDS in line 695.
18. Which "latter report" is referred to on line 757?
19. Line 764: all yeast share the same genetic code??? Do you mean all the relevant strains have the same gene content?

Reviewers' comments:

Reviewer #1 (Remarks to the Author):

This manuscript reports the genomic and phenotypic/metabolomic analyses of a series of beer yeast, specifically stout yeasts, from the Guinness beer yeast collection. These strains were whole-genome sequenced (all with short-reads [Illumina] and one with long-reads [Nanopore]), and the data used to provide insights into genealogy, phylogeny, populations structure, CNVs and SNPs of relevant genes in beer production. Phenotypic analyses via sporulation, flocculation and fermentation performance along with subsequent metabolite/flavour compound analyses provided insights into the yeasts used for the production of these highly popular beers. The manuscript report these Guinness yeasts as a unique sub-clade within the well-known Beer 1 beer yeast clade. It also reports specific, but unique domestication traits for these yeasts. Interestingly, the range of yeasts from the historical culture collection and associated records allowed the authors to track genealogy; this resource is invaluable and the authors provided some interesting insights in their analyses. To my knowledge, this is the first report of its kind on these specific beer yeasts. In my opinion, this report will attract lots of interest in the fields of beer yeast and beer yeast genomics.

The manuscript provides an interesting story with many historical insights that help in the understanding of the yeasts' development/domestication. The manuscript is interesting, the data compelling and conclusions for the most part solid. Below I provide a few comments and suggestions for the authors to address/consider.

Specific comments:

We very much thank the reviewer for taking the time to review our paper. We have addressed their comments below in green.

1. Please confirm the names of the software used in Lines 213 (“10FastSTRUCTURE”) and 217 (“11PLINK”) as accurate.

Thank you. We have corrected the typo.

2. The names of the yeast strains listed in Table 1 does not align with those used in Supplemental Data 1. Please align the latter with the former to allow easy and accurate tracking by the reader.

This has been corrected

3. Line 381-382: “Varying the number of ancestral 382 populations (K) between 1 and 10, K = 8 was found to be optimal (Figure 1).” I could not find the population analyses (K) in Figure 1. PCA, yes, but not evidence of K = 8. Please add this analysis as Fig. 1C or Supplemental Information.

We have included the additional information in Figure 1. In hindsight it was a missing and we should have included this data in our original submission; we thank the reviewer.

4. Lines 441-444. As all the other metabolite/flavour compound data are supplied in a single bar graph, it would be more useful and easy to compare the data if all the ethanol data can be reported in a single bar graph. Also, while the provided statistical analyses proved confirms the ethanol production is statistically significant, it would be more useful to provide statistical data that comment on strain-to-strain variations. I believe such analyses would be beneficial to the manuscript and might provide/support further insights. The same comment applies to the metabolites reported in Figures 4B-D.

Figure 4 amended to include ethanol production in a graph bar format. Additional analysis of strain to strain ethanol production variability reported in line 472 to 480.

5. Line 456: The authors spend quite some time discussing maltotriose metabolism. Did the authors determine the maltotriose consumption profiles or at least the end-point values to provide the relevant phenotypic link?

Analysis updated and included in figure 4

6. Line 457: multigene loci?

Amended in the text to Locus, line 490.

7. Line 489: According to Fig 4B, more than just the three stated strains surpassed 10 ppm for ethyl acetate. Please clarify.

Amended line 532

8. Line 526-530. The authors should reference their data as support; no reference currently exists. Furthermore, Figure 4D is not labelled as 2,3 butanedione (diacetyl), and in the legend 2,3 butanedione (diacetyl) is mislabeled as "C."

The text has been amended, line 562 to include reference to the figure and the typo in the figure has been amended.

9. Lines 533-535: What are these SNPs? Show the data. Are they projected to be homozygous or heterozygous? This information was supplied for the PAD1/FDC1 genes in this manuscript, and should also be supplied for all genes where SNPs are reported.

This missing has been corrected with the zygoty of the SNPs detailed in the supplementary data S5

10. Line 580-583: “The Guinness yeast, ...” FES3 seems to be above 400 ppb and the remaining production yeasts seem to be within the fairly broad 200-400 ppb threshold range. Did the authors only consider the upper limit as the threshold? Similarly, does 1955 also fall below the threshold? Please clarify.

From our internal assessment, flavour appreciation of 4 vinyl guaiacol (4VG) is very much taster orientated. We have professional tasters that will detect 4VG at <200ppb and likewise we have others that struggle <300ppb. Our assessment is commensurate with the broad range of flavour threshold reported in the literature. Additionally, our non-professional taste panel, typically brewers that have been trained to taste but are not professionals, also struggle with 4VG detection even at much higher concentrations than 400ppb. Consequently for the paper we used the 200-400ppb flavour threshold as we felt it was appropriate given the literature and our understanding of 4VG in Guinness. The 1955 yeast produces 227ppb (SD 59) of 4VG. This is just over the perceived flavour threshold although it is very much borderline, so the yeast was not included in those that are perceived to be 4VG negative.

11. Lines 589-593: The authors should reference their supporting data.

The data is now uploaded and the reference included in the manuscript. Line 1124

Illumina and Nanopore (basecalled, demultiplexed) reads for all sequenced samples in this manuscript are deposited in the European Nucleotide Archive (ENA) under the project accession PRJEB62101.

12. Lines 635-640: Show the data.

The data is now included as a supplementary figure, S7

13. Lines 712-714: Do the data generated and reported here eliminate this possibility? What about BAT2 and ILV2 and its respective CNV/metabolite correlations described in the results? While it might not provide conclusive supporting data, it cannot be dismissed either.

We thank the reviewer for their comments as CNV effect on phenotype is something that we have considered greatly. We have reviewed the data again and our conclusions in the paper that the data generated establishes that CNV does not relate to increase in metabolite production for the Guinness yeast we still countenance. However, the issue of phenotypic differences between the Guinness yeasts even though they are genotypically similar, is not resolved through this publication. This difference is of great interest not only to ourselves but we also feel that it would be beneficial to the brewing community to understand these differences as well. We hope to address these issues and resolve the CNV and phenotypic affect with further studies. We feel that the BAT2 and ILV2 CNV and phenotypic effect is too tentative to be accurate.

14. Line 722: “higher alcohols”; This statement is not entirely consistent with the description of the data in the results - see lines 508-512. This description does not eliminate the potential of CNVs in higher alcohol production. And “diacetyl”; This statement contradicts the described results - see lines 535-538.

I can see reading the comments that the lines from 508 -512 are confusing to the reader and I have tried to address this line 548 -558. In conclusion the IDS 2 and Park Royal yeast produce the same amount of Isobutanol, (t.test 0.61) but have different CNVs of *BAT2* gene, supporting the conclusion that CNV does not relate to increased phenotypic behaviour. The Diacetyl concentration, we concluded is a consequence of fermentation parameters as opposed to CNV.

Reviewer #2 (Remarks to the Author):

This is a thorough genomic and phenotypic analysis of yeasts used historically and currently in the Guinness brewery in comparison to other Irish brewing yeasts and yeasts from around the world. The main conclusion that the Guinness yeasts from a unique clade within the 'beer 1' clade of yeasts is well supported.

We very much thank the reviewer for taking the time to review our paper. We have addressed their comments below in green.

Given the close genetic relationship of the Guinness yeast strains, I find it surprising how diverse they are phenotypically. CNV does not appear to be correlated but some further general description of potential correlates of sequence variation would be helpful. What does this say about phenotypic stability - an important aspect to any industry producing a consistent product?

The observation of phenotypic diversity within the Guinness yeast is unexpected. Unlike other brewing groups that use a single pure culture, Guinness sought to mitigate the differences of a single yeast by selecting '50' colonies. This may explain why there are subtle phenotypic characteristics but not why yeast that are genotypically similar have such differences. It is something that the authors considered and is an observation we want to explore with additional projects. For us as brewers understanding the phenotype of the yeast has an important commercial aspect as well as a scientific interest.

In terms of the re-pitching and selection regimes being similar to ALE experiments it would be useful to parse the SNPs determined in the phylogeny generated - particularly the pre and post-1959 subgroups as this would help understand how genetic variation changed over time and procedures. No real discussion of loss of heterozygosity as one process of

divergence between strain lineages is had which could be part of the evolution of these strains.

We did consider the loss of heterozygosity as potential reason for the evolution of the Guinness yeast strains, however our analysis did not find any regions in the post 1959 yeast where there was a loss of heterozygosity. Additionally, the evaluation of the SNPs pre and post 1959 did not present any heterozygosity loss.

A highly detailed analysis that will be of interest to brewing yeast researchers with some interesting highlights for the more general yeast researcher.

Reviewer #3 (Remarks to the Author):

As a fan of stout, I have the pleasure of reviewing Kerruish et al titled "The Origins of the Guinness Stout Yeast".

The authors sequenced a total of 22 (13 Guinness) strains, conducted phylogenomics and phenotyping analyses. The results of manuscript is heavily biased towards the phenotyping but the histories of the strains in my opinion deserved far more attention but unfortunately was somewhat descriptive. I will focus my review on this particular section.

We very much thank the reviewer for taking the time to review our paper. We have addressed their comments below in green.

Major comment: Results on phylogeny and population structure were way too descriptive

- L382: Varying the number of ancestral populations (K) between 1 and 10, K = 8 was found to be optimal (Figure 1);

There's no information on K=8 being listed on Figure 1. Figure 1a is phylogeny and 1b is a PCA figure.

We thank the reviewer for the comment and in hindsight we should have included our analysis in the first submission. This has been addressed and is now included in analysis in figure 1

- A supplementary figure should be provided on choosing K=8 (by showing delta of different K). .

Included in Figure 1

- As the authors pointed out it "was an unexpected observation" that

"The non-Guinness Irish brewing strains were located in the 'Britain' subpopulations, whereas, the Guinness yeast occupy their own subgroup outside the USA and 'Britain' subpopulations."

The authors need to show the the Distruct plot to visualise the expected admixture proportions.

(<https://rajanil.github.io/fastStructure/>). It is difficult to pinpoint out the exact scenario from text L385-390 .

We have included our analysis of admixture in supplementary figure S2

- L388 10 yeasts were designated as being mosaic. Are these the Guinness strains? Please clarify

Mosaicism within the context of the paper is described as having <80% SNPs from a defined geographical location. We have amended Figure 1 to highlight the yeast that are mosaic.

- L496 How is this "20,000" SNPs chosen?

'The analysis of all SNPs identified in protein coding genes resulted in assessment of 20,000 protein coding biallelic SNPs' has been included in line 433

It is a real cliffhanger simply saying "Consequently It can be concluded that the Guinness strains are Mosaic"

There are so much potential with this data, and I suggest authors to perform two additional analyses:

1. Confirm source of "mosaic" / introgression by assembly and identifying the origin of phased subgenomes

(like <https://www.sciencedirect.com/science/article/pii/S0960982222001300>). Or

alternatively, simply check

where the genomes are originated from each lineage using TreeMix

(like Fig 3 <https://genome.cshlp.org/content/32/5/864>)

The referee points to these new publications that were not in press when we did our original analysis, and we thank the referee for bring this to our attention. We see this work as the start of a series of publications and hope to do further analysis on the Guinness yeast using the methodology detailed in these publications, as the referee highlights there is so much potential with this data that we feel we are only just starting on this journey.

2. Calibrate the phylogeny of Figure 2 to see pinpoint historical events of Guinness strains.

See Fig 6 of

<https://www.nature.com/articles/s41559-019-0997-9> and Fig 3 of

<https://www.nature.com/articles/s41586-020-2889-1>.

And relate the time to source in Table 1. This will greatly improve the scope of discussion at

L656-668,
boost the impact and scientific interest of this manuscript.

These publications were of great interest to us; we reviewed the methods to determine if the analysis would be appropriate for our publication. These publications assess the phylogeny of hybrid yeast and we concluded that this type of analysis is not appropriate for our project. Interestingly in our data set we discovered 3 hybrid yeast within the historical Irish brewing yeast. We are planning on exploring this data set in further publications.

Reviewer #4 (Remarks to the Author):

This paper facilitates the placement of Irish beer yeasts in the phylogeny of industrial yeasts, probably for the first time. The authors show that historical Irish beer yeasts (Smithwicks, Macardles, Perry etc.) and Guinness yeasts belong to the “Beer 1” clade, and more particularly to the “British ales” lineage, which contains isolates from the UK, the US and Ireland, as well as some likely mosaics. This is not particularly surprising. However, the authors also suggest that whereas most of the historical Irish brewing strains are closely related to British strains, the Guinness isolates are “mosaics”. Unfortunately, “mosaic” is not well defined, and the supporting data (Structure analysis) is not provided. It is therefore not clear which lineages contributed to the mosaic pattern. There are several other interesting observations. For example, unlike most other “British ale” yeasts, some Guinness isolates have acquired loss of function mutations in a maltotriose transporter, and all Guinness isolates have retained functional FDC1 and PAD1 genes, associated with off flavor (4-vinyl guaiacol) production. I cannot comment on the fermentation assays, which are outside my area of expertise.

The analysis is potentially interesting, but some data is missing, and the mosaic status of the Guinness strains should be explored in more detail. An improved manuscript will be of interest to the general community. The data will also be very useful for the growing field studying the origins and applications of industrial yeasts.

We very much thank the reviewer for taking the time to review our paper. We have addressed their comments below in green.

Major points

1. In the introduction, refer to the analysis showing that most beer yeasts result from an admixture of European and Asian wine populations (J.C Fay et al, PLoS Biol 17:e3000147 and Saada et al, Curr Biol. 32:1350).

We thank the reviewer for the comment and have included the references in line 52 and 53

2. Please improve Fig. 1A, using larger colored dots to make it possible to easily identify the geographical origin of the relevant isolates in Beer 1. Label the mosaics more clearly, and do

not include “mosaic” in the same scheme as geographical origin. Can some support values be shown, at least for the Guinness grouping and related lineages? Label the point that divides “British/US/Irish” from “German/Belgian”, and then within “British” label the point that divides “British/Irish” from “US/Guinness” (see point 3). Include the Structure analysis in Fig. 1 (see point 5).

We have amended figure 1, so that the mosaic yeast are visible within the phylogenetic tree. We have also included the FastSTRUCTURE analysis in Figure 1 and have added the admixture analysis in the supplementary figure S2.

3. Gallone et al (2016) and other studies and the current manuscript analysis show that the “Beer 1” group can be divided into two main lineages, the “German/Belgian” lineage, and the “British ales” lineage (British/Irish/US beer isolates), plus some mosaics. The exact names (“German/Belgian” or “German” and “British” or “British/Irish”) vary in different publications, but one should be chosen and used in the submitted manuscript. The current analysis shows that all the Irish yeasts (including the Guinness strains) belong to the “British ales” lineage (British/Irish/US beer isolates). Based on Fig. 1A, the “British ales” lineage can be further subdivided into two groups – one contains British isolates plus Irish isolates except for the Guinness isolates, and the second contains US isolates, Guinness isolates, and some unassigned isolates. Both groups contain some mosaic isolates. Without showing support values for the phylogeny or the PCA analysis it is difficult to know how well supported these divisions (and further subdivisions) are. The abstract and the text needs to be clearer about this point.

We have included extra analysis of the phylogeny of the Guinness yeast and these are included in figure 1 and the supplementary figure S2. The aim of this paper was to focus principally on the Guinness yeast, assessment of the other historical Irish brewing yeast was to provide a phylogenetic context for all of the Irish brewing yeast. We see this paper as the start of a wider discussion on the lineage of brewing yeast within Britain and Ireland and as such will be following up this paper with a second paper focusing upon the Irish/British lineage; we have completed the same phylogenetic analysis of the non-Guinness historical Irish brewing yeast and confirm that there are notable differences between the British and Irish lineage suggesting that phenotype of the Irish lineage is different to that reported as the British lineage.

4. I don’t follow the conclusion “The brewing strains of companies Perry, Cherry and Smithwicks aligned completely to the Britain subpopulation; whereas, the Great Northern, Macardles 1966, and Macardles 1993 yeast aligned with the ‘Britain’ group but also the US and Belgium /Germany subpopulations”. Why do you conclude that the Great Northern and Macardles align with the US or the “Belgium/German” subpopulations? This seems to be a misreading of the tree. The Guinness isolates are closer to the US isolates than Macardles from Fig. 1A and 1C, and they are all equidistant from the “Belgium/German” subpopulation. Is there evidence (e.g. from the not shown Structure analysis, see point 5) that Great Northern and Macardles isolates are mosaics? If so, please describe in more detail.

In our paper we have used the Gallone 2016 definition of Mosaic; <80% from a single geographical. Our SNP assessment of the historical Irish yeast, detailed in the table R1 below, establishes that only the Guinness yeast contain <80% SNPs from different geographical locations. Of the other historical Irish yeast; Cherry, Perry and Smithwicks align 100% to British lineage whereas the Macardles and Great northern brewery yeast contain SNPs that are aligned to Britain , Belgium/Germany and the US, however the SNPs for these three yeast aligned to Britain >80% so are not mosaic as determined by Gallone’s definition of mosaicism.

Historical Irish Yeast	Belgium/ Germany	Britain	Not Specified	Asian	Beer2	US	Wine	Mixed
Cherry	-	100%	-	-	-	-	-	-
Perry	-	100%	-	-	-	-	-	-
Smithwicks	-	100%	-	-	-	-	-	-
Great Northern	4.91%	87.36%	-	-	-	7.74%	-	-
Mcardles 1965	2.91%	89.13%	-	-	-	7.96%	-	-
Mcardles 1993	3.11%	88.59%	-	-	-	8.30%	-	-
Guinness Yeast	8%	22.40%	38.80%	-	-	30.10%	-	-

Table R1: Corresponding geographical admixture of historical Irish brewing yeast.

5. The conclusions that the Guinness yeasts are mosaics is hard to follow, partly because the Structure plot is not shown. Please show the results of the Structure analysis with K=8, which might show that the Guinness yeasts are mosaics/hybrids between a British/Irish parent and some other lineage (Belgian/German? Other lineage?). The PCA analysis does not allow the conclusion that the Guinness strains are mosaic (line 389-390). You should also clarify what you mean by “mosaic”, particularly in the context that most beer yeasts are admixed anyway (see Fay et al, point 1). Presumably the Guinness strains (and possibly BE42/58, see next point) descended from a more recent admixture.

We have included additional admixture analysis of the Guinness yeast using AlpacA (v1) software with a kmer length of 21 over 5000 base pair sliding windows; supplementary figure S2. The supplementary S2 analysis is with and without Beer042 as our analysis establish that the Guinness yeast and Beer042 are closely related. We have used the <80% SNPs from a defined geographical location so that our data set is in keeping with other previous published work (Gallone *et al.*, 2016) from which we used their data set to understand the Guinness lineage. Consequently while we note the reviewers comments about Fay *et al.*, 2019 we feel it is more in keeping with the data set to use Gallone’s definition of mosaicism.

6. State clearly that the phylogeny in Fig.1 suggests that the closest relatives of the Guinness yeasts are the mosaics/hybrids BE42 (Belgian lager yeast) and BE58 (English beer) (from Gallone et al 2016). The origin of the Guinness yeasts needs to be discussed in more depth. One interpretation is that a single hybridization occurred in the ancestor of BE42/BE58/Guinness isolates, and each lineage then diverged. It is also possible that there were multiple hybridizations. The Structure analysis should help determine if the same parent populations gave rise to all 3 lineages. Phylogenetic analysis of mosaics/hybrids can be difficult.

We thank the reviewer for this comment, as it is a question our admixture analysis further supports; relationship of Beer042 and the Guinness yeast (supplementary figure S2). We had hoped to understand the lineage of the BE42/BE58 to the Guinness yeast, and asked Prof Verstaappen for more information about these yeast. Our thought process was to then follow up historical information with further genotypic work but unfortunately Prof Verstaappen is unable to collaborate or discuss his work with other brewers due to the AB Inbev funding stipulation. This is something we hope to revisit in the future but we do not feel that we can conclude the story successfully without the historical information for context.

7. The discussion of the maltose/maltotriose gene loci is confusing. MAL loci are associated with utilization of maltose (for maltotriose, see next point). *S. cerevisiae* genomes differ in the number of MAL loci that they contain, and Beer1 isolates usually have many copies. It should be possible to determine the number and location of the loci in the Guinness IDS1 isolate from the MinION assembly. Please include this in a diagram/figure, which will greatly clarify the description. You can then determine if any of the other Guinness isolates have more/fewer loci by mapping the Illumina reads on the MinION assembly. For example, it is very unlikely that MAL63 has moved to Chr XVI. It is more likely that there is an ancestral maltose transporter on Chr XVI (like there is the reference genome) and that MAL63 has been lost.

The *MAL6* operon warrants further investigation as we also found the arrangement of the genes to be unusual, our conclusions are tentative until we have done further work which we expect to do in a follow up paper on the observation. Our MinION assessment of IDS1 and the corresponding scaffolding of the other Guinness yeast suggests that the arrangement of the *MAL6* operon, *MAL61* and *MAL62* on VIII and *MAL63* mapped to the sub telomeric region of chromosome XVI, is replicated throughout all of the Guinness yeast. We assessed the effects of ploidy and our observations were repeated through the sequencing data. Through our assessment we were unable to locate *MAL2* and *MAL4*. The sequencing data (supplementary figure S3) has a homozygous stop codon at position 1075820 in *MAL11* suggesting that this gene is potentially likely to be non-functioning although the occurrence of this stop codon is present within 145 of the 176 yeast assessed in this study. The reported frameshift mutations in *MAL31* are heterozygous and do not lead to a loss in function. Sugar analysis at the end of the fermentations, figure 2,

establishes that all the wort sugars are utilised by the Guinness yeast during fermentations. The *MAL63* arrangement conclusions remain tentative until further work is done which we hope to do with a follow up paper.

8. Most MAL loci encode maltose transporters. However, the MAL11 transporter (also called AGT1) at MAL1 is different. This also transports maltotriose. A frameshift in MAL11 would prevent transport of maltotriose, whereas maltose would still be transported if other MAL transporter genes are intact (line 463). Frameshifts in MAL11/AGT1 are indeed unusual in the “British ales” lineage. Are the frameshifts in the 4 Guinness strains homozygous? You should determine which of the Guinness strains can utilize maltotriose and which can utilize maltose, and see if this correlates with the frameshift in MAL11.

We have reviewed the data in light of the referee’s comments. We can confirm that the *MAL11* frame shift mutations described in the paper’s original submission were heterozygous, however following a review of the data we determined that there was a homozygous premature stop codon present at position 1075820 in all of the Guinness yeast, and 145 of the 176 yeast assessed in our study. The *MAL31* and *MAL61* genes encode maltotriose transporters (Alves *et al.*, 2008; Day *et al.*, 2002). An additional blast search of the Guinness yeast *MAL61* confirmed that the sequence shared a 99% (1837/1842 bp) similarity to a previously described maltotriose utilising gene MTT1 (Dietvorst *et al.*, 2005). Additionally the lack of maltotriose at the end of fermentation (figure 2) further supports our contention that the Guinness yeast utilise maltotriose.

9. The discussion of off flavors (4-vinyl guaiacol) is interesting. Most Beer 1 yeasts (and all “British ales” yeasts) have frameshifts in FDC1 and PAD1 required to make 4VG (Gallone *et al* 2016, Goncalves *et al* 2016, Saada *et al* 2022), as noted in the discussion. One question is therefore where did the intact genes in the Guinness isolates come from? Were they retained from the ancestor of “British ales” yeasts, and acquired frameshifts in every lineage except for the Guinness isolates, and the wheat beers? My interpretation of supplementary material in Gallone *et al* is that in BE42 and BE58 (closest relatives of the Guinness yeasts) FDC1 and PAD1 do contain loss-of-function mutations, though in one isolate there are heterozygous mutations. This would suggest that FDC1/PAD1 may have been fully functional in the immediate ancestor of BE42/BE58/Guinness. The functional copies could have originated from a Belgian/German parent, where some isolates have intact FDC1 and PAD1 genes (Saada *et al.* 2022). This is probably worth discussing. Can you indicate the position of the wheat beer isolates on your phylogeny?

We didn’t comment upon BE42 and BE58 for the reasons discussed in reviewers question 6. The wheat beer isolates are highlighted in the supplementary data sheet S6.

10. I can’t find a reference to availability of the raw data, though I may have missed it in one of the online documents. The raw data must be available, and the MinION assembly and the annotation must also be provided (if they were available I could look at the MAL loci for example). The custom script (line 222) should also be provided.

The data is now uploaded and the reference included in the manuscript. Line 1124

Illumina and Nanopore (basecalled, demultiplexed) reads for all sequenced samples in this manuscript are deposited in the European Nucleotide Archive (ENA) under the project accession PRJEB62101.

Minor -easily addressed

11. The genome data from Smithwicks, Macardles etc. is underused and little background is provided. Could more information about Perry, Cherry, Smithwicks, McCardles etc. breweries be provided? Do these isolates have mutations in MAL1, FDC1, PAD1 etc? Did one of these lineages contribute to the mosaic structure of the Guinness strains?

We are planning to publish a follow up paper detailing the different phenotypes of the other non Guinness historical Irish brewing yeast. All of the non Guinness breweries and Guinness were members of the Irish Ale Breweries association (estd 1962). The association shared technical information and yeast, so the different lineages established in this first paper were a surprise to us. In addition our phenotypic assessment of the non-Guinness Irish historical yeast suggests that there are phenotypes within the Irish population of the 'British' grouping that are different from the other British yeast .

12. The methods for how the phylogeny was constructed are not provided. A phylogeny cannot be constructed using SnpEff (line 364). The legend of Fig. 1 refers to Maximum Likelihood, but more information is required.

The omission of the relevant information was incorrect and we thank the reviewer for their comment in highlighting this inaccuracy . We have now amended figure 1 and in the text stating which programmes have been used and the relevant scientific procedure.

13. One note on using Flye for assembly – this can introduce errors, especially with polyploid eukaryotic genomes. Canu (PMID: 28298431) is often a better choice. You may even have enough raw reads to generate a phased assembly, even for parts of the genome. This would help to determine the origins of the parents of the Guinness strains. (This is a suggestion, not a requirement).

We thank the reviewer for their comment and will explore the use of this programme in potential future works

14. Please clarify the number of strains used, which varies from 19 to 22, depending on whether "IDS1" and "IDS2" etc are assumed to be the same or not (e.g. line 116). Make it clear in Table 1 that "Park Royal" are Guinness isolates.

It was customary for Guinness not to use a completely pure yeast strain, instead they pooled the yeast that were genotypically and phenotypically very similar. This was at the time the accepted procedure for ale and stout brewers with many ale yeast cultures being a

mix of yeast. This is contrary to lager brewers who have chosen to use a pure yeast strain of *S. pastorianus*. As an industry we have moved to the pure yeast strain, but due to the historical legacy of the Guinness yeast within the mater culture there are these different variations. This explains the differences in IDS1 and IDS2, subsequently we feel that these yeast should be considered as separate and have been included in the paper as such. We have amended table 1 so that the reader finds it easier to follow.

15. Aneuploidy (Figure 3): there is a difference in the copy number of Chr X in IDS1 and IDS2, which should be commented. You may wish to check Chr XV in FES2. It is surprising that part of this is colored black.

We thank the reviewer and have rechecked our data, the observations of the CNV reported in the paper are correct. We have made a comment in line 451 discussing the different CNV in IDS1 and IDS2.

16. CNVs: did you account for aneuploidy when calculating copy number? BAT2 is on Chr X, and there are extra copies of Chr X in PR1960. The section on CNV and gene content (from around line 504) is somewhat long and speculative.

The analysis in CNV is calculated accounting for the aneuploidy of the yeast. The analysis on the BAT2 was re-evaluated as requested by the reviewer and have confirmed that our original observations are correct. We have amended the manuscript, lines 555– 561.

17. I can't follow the description of 1981 IDS in line 695.

Have amended please see line 747.

18. Which "latter report" is referred to on line 757?

Have included the reference omitting the 'latter report' statement. Line 808

19. Line 764: all yeast share the same genetic code??? Do you mean all the relevant strains have the same gene content?

Have amended, line 815.

reference

Alves Jr, S.L., Herberts, R.A., Hollatz, C., Trichez, D., Miletti, L.C., de Araujo, P.S. and Stambuk, B.U., 2008. Molecular analysis of maltotriose active transport and fermentation by *Saccharomyces cerevisiae* reveals a determinant role for the AGT1 permease. *Applied and environmental microbiology*, 74(5), pp.1494-1501.

Day, R.E., Rogers, P.J., Dawes, I.W. and Higgins, V.J., 2002. Molecular analysis of maltotriose transport and utilization by *Saccharomyces cerevisiae*. *Applied and environmental microbiology*, 68(11), pp.5326-5335.

Dietvorst, J., Londesborough, J. and Steensma, H.Y., 2005. Maltotriose utilization in lager yeast strains: MTT1 encodes a maltotriose transporter. *Yeast*, 22(10), pp.775-788.

** See the Nature Portfolio author and referees' website at www.nature.com/authors for information about policies, services and author benefits

Communications Biology is committed to improving transparency in authorship. As part of our efforts in this direction, we are now requesting that all authors identified as 'corresponding author' create and link their Open Researcher and Contributor Identifier (ORCID) with their account on the Manuscript Tracking System prior to acceptance. ORCID helps the scientific community achieve unambiguous attribution of all scholarly contributions. You can create and link your ORCID from the home page of the Manuscript Tracking System by clicking on 'Modify my Springer Nature account' and following the instructions in the link below. Please also inform all co-authors that they can add their ORCIDs to their accounts and that they must do so prior to acceptance.

If you experience problems in linking your ORCID, please contact the Platform Support Helpdesk.

This email has been sent through the Springer Nature Tracking System NY-610A-NPG&MTS

Confidentiality Statement:

This e-mail is confidential and subject to copyright. Any unauthorised use or disclosure of its contents is prohibited. If you have received this email in error please notify our Manuscript Tracking System Helpdesk team at <http://platformsupport.nature.com>.

Details of the confidentiality and pre-publicity policy may be found here <http://www.nature.com/authors/policies/confidentiality.html>

Privacy Policy | Update Profile

Reviewers' comments:

Reviewer #1 (Remarks to the Author):

This is a review of the revised submission of the original manuscript; a general summary of the goals, results and conclusions of the manuscript would be repetitive. I find the authors adequately addressed the scientific concerns raised in the first review. That said, the authors can consider the following general comments and suggestions to further improve/clarify the manuscript.

1. L275-287. What was the internal standard used for determining the concentration of 4-VG in the beer. According to L281 it was 4-VG.
2. In general, the labels of the figures are too small and unclear.
3. Supplemental Table 7: This Excel file contains three spreadsheets of which 2 contributes nothing to the manuscript. The authors should consider deleting the last two spreadsheets (FLO8 and FLO11) and renaming the first (FLO9) to encompass all the genes represented.
4. Several manuscript formatting inconsistencies exist; these include: (a) inconsistent use of number-unit nomenclature (e.g., cells/mL vs cells mL⁻¹; and rates as x/min vs x min⁻¹. (b) Periods are present/absent at the ends of subheadings. (c) The use of the abbreviated "S. cerevisiae" is inconsistent. (d) Inconsistent reporting of statistical results (e.g., t-test - L556-558). The journal formatting guide will help eliminate these inconsistencies.
5. Several grammatical inconsistencies exist. These include, but are not limited to: (a) Periods are often absent at the ends of sentences; (b) Apostrophes are unnecessarily used (e.g., brewers rather than brewers'); (c) Several spelling errors are present (e.g., bottle conditioned not bottled conditioned; Beer 1 clade not beer 1 clade; Hefeweizen not hefeweizen; S288c and s288c are used interchangeably; correlation not corelation). (d) Some convoluted sentences exist (e.g., L473; L642).

Again, I find the paper interesting and it will likely attract wide interest from brewers and researchers alike.

Reviewer #2 (Remarks to the Author):

I am happy with the revised version.

Reviewer #3 (Remarks to the Author):

The authors have dealt my previous comments adequately.

[In response to Reviewer #4's comments]

Having read the manuscript again, I recommend the authors to write more objectively on the potential evolutionary scenario of Guinness strains. We all know Guinness tastes great and any findings presented in this paper will not boost or downplay the industry. This is what I urge the authors to think about.

With this in mind, let's refer to point 5 and 6.

I think there's a lot of details missing in the population structure analyses and one should discuss more extensively. It seems that Figure 1C that some strains (n= 161) are omitted (n= 176 in Figure 1A and 1B). Please be consistent explain in figure legend or methods.

Regarding

L394. ... $K = 8$ was found to be optimal

L412-417 Consequently, 412 analysis present in supplementary figure 2 confirms the highly non-linear genetic content of 413 the Guinness yeast relative to other members of the Beer1 clade. Furthermore the 414 phylogenetic tree presented in Figure 1 placed the yeast Beer042 as the closest relative to 415 the Guinness yeast. The Alpaca analysis confirms this observation with all of the 416 chromosomes sharing SNPs observed in Beer042. Omission of the Beer042 from the analysis 417 confirmed that SNPs from the 'not specified' ancestry were the most significant (38.8%) 418 (supplementary figure S2).

Hence, Beer042 should be clearly labelled in Figure 1C. And the findings or analyses should be evolved around the fact that the Guinness strains are i) grouped from strain Beer042. Could the Beer042 be an 'ancestor strain' of the Guinness strains, or it's a Guinness strain that have undergone introgression with other Belgian mixed strains (hence being placed 'basal' in the phylogeny) ?

Simply omit Beer042 (L412-L417) from the analyses is not straight forward and the authors should directly mention this in Result and Discussion. It is OKAY if Prof Verstappen decide not to share the 'followup information', because the current information that the Guinness strains group with this lineage was already fascinating.

I urge the authors to redo this part of analyses, make everything more consistent. It is in my opinion more straight forward to discuss this way rather than showing the "not specified" ancestry constitute ~40% of Guinness strains. Then it really did not reveal anything. You should discuss on what's currently known, which is already fascinating.

More minor comments

L431 Flye not Flybe

All de novo should be italicised.

Please check typos throughout.

Reviewers' comments:

The authors thank the reviewers for taking the time to review our publication, The origins of the Guinness Stout Yeast. We have addressed the reviewers' comments in the resubmitted paper and have responded to the remarks below in green. The comments of the reviewers have in our opinion greatly improved the paper, which we thank them for. We appreciate that reviewing papers are a significant request, and again thank the reviewers for their invaluable input.

Reviewer #1 (Remarks to the Author):

This manuscript reports the genomic and phenotypic/metabolomic analyses of a series of beer yeast, specifically stout yeasts, from the Guinness beer yeast collection. These strains were whole-genome sequenced (all with short-reads [Illumina] and one with long-reads [Nanopore]), and the data used to provide insights into genealogy, phylogeny, populations structure, CNVs and SNPs of relevant genes in beer production. Phenotypic analyses via sporulation, flocculation and fermentation performance along with subsequent metabolite/flavour compound analyses provided insights into the yeasts used for the production of these highly popular beers. The manuscript report these Guinness yeasts as a unique sub-clade within the well-known Beer 1 beer yeast clade. It also reports specific, but unique domestication traits for these yeasts. Interestingly, the range of yeasts from the historical culture collection and associated records allowed the authors to track genealogy; this resource is invaluable and the authors provided some interesting insights in their analyses. To my knowledge, this is the first report of its kind on these specific beer yeasts. In my opinion, this report will attract lots of interest in the fields of beer yeast and beer yeast genomics.

The manuscript provides an interesting story with many historical insights that help in the understanding of the yeasts' development/domestication. The manuscript is interesting, the data compelling and conclusions for the most part solid. Below I provide a few comments and suggestions for the authors to address/consider.

Specific comments:

1. Please confirm the names of the software used in Lines 213 ("10FastSTRUCTURE") and 217 ("11PLINK") as accurate.

Thank you. We have corrected the typo. Line 216 and 221.

2. The names of the yeast strains listed in Table 1 does not align with those used in Supplemental Data 1. Please align the latter with the former to allow easy and accurate tracking by the reader.

This has been corrected

3. Line 381-382: "Varying the number of ancestral 382 populations (K) between 1 and 10, K = 8 was found to be optimal (Figure 1)." I could not find the population analyses (K) in Figure 1. PCA, yes, but not evidence of K = 8. Please add this analysis as Fig. 1C or Supplemental Information.

We have included the additional information in Figure 1. In hindsight it was a missing and we should have included this data in our original submission; we thank the reviewer.

4. Lines 441-444. As all the other metabolite/flavour compound data are supplied in a single bar graph, it would be more useful and easy to compare the data if all the ethanol data can be reported in a single bar graph. Also, while the provided statistical analyses proved confirms the ethanol production is statistically significant, it would be more useful to provide statistical data that comment on strain-to-strain variations. I believe such analyses would be beneficial to the manuscript and might provide/support further insights. The same comment applies to the metabolites reported in Figures 4B-D.

Figure 4 amended to include ethanol production in a graph bar format. Additional analysis of strain to strain ethanol production variability reported in line 472 to 480.

5. Line 456: The authors spend quite some time discussing maltotriose metabolism. Did the authors determine the maltotriose consumption profiles or at least the end-point values to provide the relevant phenotypic link?

Analysis updated and included in figure 4

6. Line 457: multigene loci?

Amended in the text to Locus, line 492.

7. Line 489: According to Fig 4B, more than just the three stated strains surpassed 10 ppm for ethyl acetate. Please clarify.

Amended line 533

8. Line 526-530. The authors should reference their data as support; no reference currently exists. Furthermore, Figure 4D is not labelled as 2,3 butanedione (diacetyl), and in the legend 2,3 butanedione (diacetyl) is mislabeled as "C."

The text has been amended, line 573 to include reference to the figure and the typo in the figure has been amended.

9. Lines 533-535: What are these SNPs? Show the data. Are they projected to be

homozygous or heterozygous? This information was supplied for the PAD1/FDC1 genes in this manuscript, and should also be supplied for all genes where SNPs are reported.

This missing has been corrected with the zygosity of the SNPs detailed in the supplementary data S5

10. Line 580-583: "The Guinness yeast, ..." FES3 seems to be above 400 ppb and the remaining production yeasts seem to be within the fairly broad 200-400 ppb threshold range. Did the authors only consider the upper limit as the threshold? Similarly, does 1955 also fall below the threshold? Please clarify.

From our internal assessment, flavour appreciation of 4 vinyl guaiacol (4Vg) is very much taster orientated. We have professional tasters that will detect 4VG at <200ppb and likewise we have others that struggle <300ppb. Our assessment is commensurate with the broad range of flavour threshold reported in the literature. Additionally, our non-professional taste panel, typically brewers that have been trained to taste but are not professionals, also struggle with 4VG detection even at much higher concentrations than 400ppb. Consequently, for the paper we used the 200-400ppb flavour threshold as we felt it was appropriate given the literature and our understanding of 4VG in Guinness. The 1955 yeast produces 227ppb (SD 59) of 4VG. This is just over the perceived flavour threshold although it is very much borderline, so the yeast was not included in those that are perceived to be 4VG negative.

11. Lines 589-593: The authors should reference their supporting data.

The data is now uploaded and the reference included in the manuscript. Line 1169

Illumina and Nanopore (basecalled, demultiplexed) reads for all sequenced samples in this manuscript are deposited in the European Nucleotide Archive (ENA) under the project accession PRJEB62101.

12. Lines 635-640: Show the data.

The data is now included as a supplementary figure, S7

13. Lines 712-714: Do the data generated and reported here eliminate this possibility? What about BAT2 and ILV2 and its respective CNV/metabolite correlations described in the results? While it might not provide conclusive supporting data, it cannot be dismissed either.

We thank the reviewer for their comments as CNV effect on phenotype is something that we have considered greatly. We have reviewed the data again and our conclusions in the paper that the data generated establishes that CNV does not relate to increase in metabolite production for the Guinness yeast we still countenance. However, the issue of phenotypic differences between the Guinness yeasts even though they are genotypically

similar, is not resolved through this publication. We have added additional comment on lines 779- 811 discussing genotype and phenotype. The difference in phenotype and genotype is of great interest not only to ourselves but we also feel that it would be beneficial to the brewing community to understand these differences as well. We hope to address these issues and resolve the CNV and phenotypic affect with further studies. We feel that the BAT2 and ILV2 CNV and phenotypic effect is too tentative to be accurate.

14. Line 722: “higher alcohols”; This statement is not entirely consistent with the description of the data in the results - see lines 508-512. This description does not eliminate the potential of CNVs in higher alcohol production. And “diacetyl”; This statement contradicts the described results - see lines 535-538.

I can see reading the comments that the lines from 508 -512 are confusing to the reader and I have tried to address this line 548 -558. In conclusion the IDS 2 and Park Royal yeast produce the same amount of Isobutanol, (t.test 0.61) but have different CNVs of *BAT2* gene, supporting the conclusion that CNV does not relate to increased phenotypic behaviour. The Diacetyl concentration, we concluded is a consequence of fermentation parameters as opposed to CNV.

Reviewer #2 (Remarks to the Author):

This is a thorough genomic and phenotypic analysis of yeasts used historically and currently in the Guinness brewery in comparison to other Irish brewing yeasts and yeasts from around the world. The main conclusion that the Guinness yeasts from a unique clade within the 'beer 1' clade of yeasts is well supported.

Given the close genetic relationship of the Guinness yeast strains, I find it surprising how diverse they are phenotypically. CNV does not appear to be correlated but some further general description of potential correlates of sequence variation would be helpful. What does this say about phenotypic stability - an important aspect to any industry producing a consistent product?

The difference in phenotype of the Guinness yeast is of great interest to us and we have added additional comment on lines 779- 811 discussing genotype and phenotype.

In terms of the re-pitching and selection regimes being similar to ALE experiments it would be useful to parse the SNPs determined in the phylogeny generated - particularly the pre and post-1959 subgroups as this would help understand how genetic variation changed over time and procedures. No real discussion of loss of heterozygosity as one process of divergence between strain lineages is had which could be part of the evolution of these strains.

We did an initial review of the loss of heterozygosity (LOH) but revisited LOH in greater detail following the reviewers comments, which we thank them for. The LOH of the Guinness yeast are similar to other yeast used in beer production (10-15%). The recent publication by Peirs *et al.*, 2023 further corroborates our analysis as *S. cerevisiae* have higher levels of heterozygosity than other *Saccharomyces* species which the authors conclude is a consequence of domestication. The reason we did not include the LOH analysis of the Guinness yeast in the manuscript is that the LOH did not differentiate upon lineage. Our analysis did not find any specific regions in the post 1959 yeast where there was a loss of heterozygosity compared to the pre 1959 Guinness yeast. Subsequently we did not include this submission in the manuscript as LOH cannot explain the observed divergence between the pre and post 1959 samples nor the difference in phenotypic behaviour.

A highly detailed analysis that will be of interest to brewing yeast researchers with some interesting highlights for the more general yeast researcher.

Reviewer #3 (Remarks to the Author):

As a fan of stout, I have the pleasure of reviewing Kerruish et al titled "The Origins of the Guinness Stout Yeast".

The authors sequenced a total of 22 (13 Guinness) strains, conducted phylogenomics and phenotyping analyses. The results of manuscript is heavily biased towards the phenotyping but the histories of the strains in my opinion deserved far more attention but unfortunately was somewhat descriptive. I will focus my review on this particular section.

Major comment: Results on phylogeny and population structure were way too descriptive

- L382: Varying the number of ancestral populations (K) between 1 and 10, K = 8 was found to be optimal (Figure 1);

There's no information on K=8 being listed on Figure 1. Figure 1a is phylogeny and 1b is a PCA figure.

We thank the reviewer for the comment and in hindsight we should have included our analysis in the first submission. This has been addressed and is now included in analysis in

figure 1

- A supplementary figure should be provided on choosing K=8 (by showing delta of different K). .

Included in Figure 1

- As the authors pointed out it "was an unexpected observation" that "The non-Guinness Irish brewing strains were located in the 'Britain' subpopulations, whereas, the Guinness yeast occupy their own subgroup outside the USA and 'Britain' subpopulations."

The authors need to show the the Distruct plot to visualise the expected admixture proportions.

(<https://rajanil.github.io/fastStructure/>). It is difficult to pinpoint out the exact scenario from text L385-390 .

We have included our analysis of admixture in supplementary figure S2

- L388 10 yeasts were designated as being mosaic. Are these the Guinness strains? Please clarify

Mosaicism within the context of the paper is described as having <80% SNPs from a defined geographical location. We have amended Figure 1 to highlight the yeast that are mosaic.

- L496 How is this "20,000" SNPs chosen?

The SNPs were chosen as they were identified in protein coding genes. This type of analysis had been previously used by Peters *et al.*, 2018 to establish *S. cerevisiae* phylogeny. We have updated the manuscript line 435.

It is a real cliffhanger simply saying "Consequently It can be concluded that the Guinness strains are Mosaic"

There are so much potential with this data, and I suggest authors to perform two additional analyses:

1. Confirm source of "mosaic" / introgression by assembly and identifying the origin of phased subgenomes

(like <https://www.sciencedirect.com/science/article/pii/S0960982222001300>). Or alternatively, simply check

where the genomes are originated from each lineage using TreeMix

(like Fig 3 <https://genome.cshlp.org/content/32/5/864>)

We very much thank the reviewer for this comment as it made us question which analysis would provide the best insight as to the recent lineage of the Guinness yeast. Before resubmitting the paper the authors discussed at length different analysis and we decided to

use Alpac software to determine the likely origin of the Guinness yeast using continuous chromosomal segments. The reason for this choice was that the work by Saada *et al.*, 2022 and Fay *et al.* 2019 establish that all of the *S. cerevisiae* beer 1 yeast are of European and Asian admixture. Our analysis establishes that the Guinness yeast groups within the beer 1 clade. Consequently, the SNPs reported in the Guinness yeast would align with the work described by Saada *et al.*, 2022 and Fay *et al.* 2019 that the origins of the Guinness yeast SNPs are of the European and Asian admixture. What we wanted to do is understand the recent lineage of the Guinness yeast allowing understanding where these more recent SNPs have originated from, thus providing insight as to how the Guinness yeast came into being.

We decided to use Alpac software to determine the likely origin of the continuous chromosomal segments using a kmer based approach over 5000 base pair sliding windows across the entire genome. This analysis complements the admixture based approach in suggesting a non-linear monophyletic origin of the Guinness strains. This analysis present in supplementary figure 2 firmly establishes the highly non-linear genetic content of the Guinness strains relative to other members of the Beer1 clade. The data demonstrates that non-overlapping segments across all chromosomes within the Guinness yeast display a high degree of similarity with segments from yeasts derived from very distinct phylogenetic subclades and geographical locations. The Guinness yeast share a very distinct lineage with the previously described Beer042 yeast; this relationship between Beer042 and the Guinness yeast is so close that we present a with and without Beer042 in supplementary figure 2.

Our assessment of the Guinness yeast establishes the mosaic nature of the Guinness yeast genome as defined by Gallone *et al.*, 2016 definition as <80% SNPs from one prescribed geographical location, and indicates the likely contribution of several established lineages to the genetic make-up of the Guinness strains. The analysis of Saada *et al.*, 2022 and Fay *et al.*, 2019 are of great interest to anyone assessing all of the *S. cerevisiae* lineage, as they were to us, however in this instance we wanted to understand the recent accumulation of SNPs for the Guinness yeast which our analysis establishes.

2. Calibrate the phylogeny of Figure 2 to see pinpoint historical events of Guinness strains.

See Fig 6 of

<https://www.nature.com/articles/s41559-019-0997-9> and Fig 3 of

<https://www.nature.com/articles/s41586-020-2889-1>.

And relate the time to source in Table 1. This will greatly improve the scope of discussion at L656-668,

boost the impact and scientific interest of this manuscript.

The additional supplementary figure, S2, in the revised manuscript details the origins of the SNPs in 5000 bp window. We have commented upon this inclusion in lines 409-419. Our conclusion is that the Guinness yeast lineage is of non-linear genetic origin, which is different from the other historical brewing yeast which are of 'British' origin. The reason for

this difference we consider to be a consequence of the type of beer produced by these yeasts; Guinness yeast principally brew stouts whereas the other Irish brewing yeast brew ales. We have commented upon this in lines 845-872.

The publications referenced by the reviewer are of great interest to us as they discuss hybrid yeast; particularly as during our analysis of historical Irish brewing yeast we discovered that two Irish brewers used a hybrid yeast of *S. cerevisiae* x *S. kudriavzevii*. Gallone *et al.*, 2019 conclude that *S. cerevisiae* x *S. kudriavzevii* hybrid belong principally to a monophyletic clade that is associated with Belgium ale brewing: Lambic and Trappist. Analysing these Irish brewing hybrid yeast could potentially aid in our understanding of hybridizing events especially if the hybridization event observed in the Irish brewing yeast is different to the Belgium *S. cerevisiae* x *S. kudriavzevii* hybrid. Introgression analysis as described by D'Angiolo *et al.*, 2020 of the *S. cerevisiae* in the *S. cerevisiae* x *S. kudriavzevii* Irish brewing hybrids would also be of interest as well, as it would provide further understanding of brewing yeast lineage in Ireland. We did not include these hybrid yeast in our Guinness stout manuscript as their inclusion merits a separate study and publication. Interestingly these hybrid yeast are from breweries located in the same city in Ireland but this city is separate from where the Guinness brewery is located.

Reviewer #4 (Remarks to the Author):

This paper facilitates the placement of Irish beer yeasts in the phylogeny of industrial yeasts, probably for the first time. The authors show that historical Irish beer yeasts (Smithwicks, Macardles, Perry etc.) and Guinness yeasts belong to the "Beer 1" clade, and more particularly to the "British ales" lineage, which contains isolates from the UK, the US and Ireland, as well as some likely mosaics. This is not particularly surprising. However, the authors also suggest that whereas most of the historical Irish brewing strains are closely related to British strains, the Guinness isolates are "mosaics". Unfortunately, "mosaic" is not well defined, and the supporting data (Structure analysis) is not provided. It is therefore not clear which lineages contributed to the mosaic pattern. There are several other interesting observations. For example, unlike most other "British ale" yeasts, some Guinness isolates have acquired loss of function mutations in a maltotriose transporter, and all Guinness isolates have retained functional FDC1 and PAD1 genes, associated with off flavor (4-vinyl guaiacol) production. I cannot comment on the fermentation assays, which are outside my area of expertise.

The analysis is potentially interesting, but some data is missing, and the mosaic status of the Guinness strains should be explored in more detail. An improved manuscript will be of interest to the general community. The data will also be very useful for the growing field studying the origins and applications of industrial yeasts.

Major points

1. In the introduction, refer to the analysis showing that most beer yeasts result from an

admixture of European and Asian wine populations (J.C Fay et al, PLoS Biol 17:e3000147 and Saada et al, Curr Biol. 32:1350).

We thank the reviewer for the comment and have included the references in line 52 and 53

2. Please improve Fig. 1A, using larger colored dots to make it possible to easily identify the geographical origin of the relevant isolates in Beer 1. Label the mosaics more clearly, and do not include “mosaic” in the same scheme as geographical origin. Can some support values be shown, at least for the Guinness grouping and related lineages? Label the point that divides “British/US/Irish” from “German/Belgian”, and then within “British” label the point that divides “British/Irish” from “US/Guinness” (see point 3). Include the Structure analysis in Fig. 1 (see point 5).

We have amended figure 1, so that the mosaic yeast are visible within the phylogenetic tree. We have also included the FastSTRUCTURE analysis in Figure 1 and have added the admixture analysis in the supplementary figure S2.

3. Gallone et al (2016) and other studies and the current manuscript analysis show that the “Beer 1” group can be divided into two main lineages, the “German/Belgian” lineage, and the “British ales” lineage (British/Irish/US beer isolates), plus some mosaics. The exact names (“German/Belgian” or “German” and “British” or “British/Irish”) vary in different publications, but one should be chosen and used in the submitted manuscript. The current analysis shows that all the Irish yeasts (including the Guinness strains) belong to the “British ales” lineage (British/Irish/US beer isolates). Based on Fig. 1A, the “British ales” lineage can be further subdivided into two groups – one contains British isolates plus Irish isolates except for the Guinness isolates, and the second contains US isolates, Guinness isolates, and some unassigned isolates. Both groups contain some mosaic isolates. Without showing support values for the phylogeny or the PCA analysis it is difficult to know how well supported these divisions (and further subdivisions) are. The abstract and the text needs to be clearer about this point.

We have included extra analysis of the phylogeny of the Guinness yeast and these are included in figure 1 and the supplementary figure S2. The aim of this paper was to focus principally on the Guinness yeast, assessment of the other historical Irish brewing yeast was to provide a phylogenetic context for all of the Irish brewing yeast. We see this paper as the start of a wider discussion on the lineage of brewing yeast within Britain and Ireland and as such will be following up this paper with a second paper focusing upon the Irish/British lineage; we have completed the same phenotypic analysis of the non-Guinness historical Irish brewing yeast and confirm that there are notable differences between the British and Irish lineage suggesting that phenotype of the Irish lineage is different to that reported as the British lineage.

4. I don't follow the conclusion “The brewing strains of companies Perry, Cherry and Smithwicks aligned completely to the Britain subpopulation; whereas, the Great Northern,

Macardles 1966, and Macardles 1993 yeast aligned with the ‘Britain’ group but also the US and Belgium /Germany subpopulations”. Why do you conclude that the Great Northern and Macardles align with the US or the “Belgium/German” subpopulations? This seems to be a misreading of the tree. The Guinness isolates are closer to the US isolates than Macardles from Fig. 1A and 1C, and they are all equidistant from the “Belgium/German” subpopulation. Is there evidence (e.g. from the not shown Structure analysis, see point 5) that Great Northern and Macardles isolates are mosaics? If so, please describe in more detail.

The omission of the population structure was in hindsight an error, and we have addressed this issue in our resubmission, please see figure 1 C. In the study we have used the Gallone *et al.*, 2016 definition of Mosaic; <80% from a single geographical. Our SNP assessment of the historical Irish yeast, detailed in the table R1 below, establishes that only the Guinness yeast contain <80% SNPs from different geographical locations. Furthermore, we have sought greater clarity on the origins of the Guinness yeast by using Alpaca analysis. Using the Alpaca (v1) software and a kmer length of 21 over 5000 base pair sliding windows we established that the Guinness yeast SNPs are of multiple origins and as such the Guinness yeast is of non-linear monophyletic origin (supplementary figure 2). This analysis confirms our assessment that the Guinness yeast are mosaic in nature. Of the other historical Irish yeast; Cherry, Perry and Smithwicks align 100% to British lineage whereas the Macardles and Great northern brewery yeast contain SNPs that are aligned to Britain , Belgium/Germany and the US, however the SNPs for these three yeast aligned to Britain >80% so are not mosaic as determined by Gallone’s *et al.*, 2016 definition of mosaicism.

Historical Irish Yeast	Belgium/ Germany	Britain	Not Specified	Asian	Beer2	US	Wine	Mixed
Cherry	-	100%	-	-	-	-	-	-
Perry	-	100%	-	-	-	-	-	-
Smithwicks	-	100%	-	-	-	-	-	-
Great Northern	4.91%	87.36%	-	-	-	7.74%	-	-
Mcardles 1965	2.91%	89.13%	-	-	-	7.96%	-	-
Mcardles 1993	3.11%	88.59%	-	-	-	8.30%	-	-
Guinness Yeast	8%	22.40%	38.80%	-	-	30.10%	-	-

Table R1: Corresponding geographical admixture of historical Irish brewing yeast.

5. The conclusions that the Guinness yeasts are mosaics is hard to follow, partly because the Structure plot is not shown. Please show the results of the Structure analysis with K=8, which might show that the Guinness yeasts are mosaics/hybrids between a British/Irish

parent and some other lineage (Belgian/German? Other lineage?). The PCA analysis does not allow the conclusion that the Guinness strains are mosaic (line 389-390). You should also clarify what you mean by “mosaic”, particularly in the context that most beer yeasts are admixed anyway (see Fay et al, point 1). Presumably the Guinness strains (and possibly BE42/58, see next point) descended from a more recent admixture.

We found the work of work of Fay *et al.*, 2019 and Saada *et al.*, 2022 of great interest as both groups concluded, as the reviewer states, that the yeast belonging to the Beer 1 clade are of European and Asian admixture. We were conscious that if we used the same analysis as Saada *et al.*, 2022 and Fay *et al.*, 2019 on the Guinness yeast the conclusion would be the similar. Whereas the Guinness yeast stout project sought to understand the recent lineage of the Guinness yeast as we want to determine how the Guinness yeast came into being. Subsequently, we chose to analysed the Guinness yeast using Alpaca (v1) software with a kmer length of 21 over 5000 base pair sliding windows; supplementary figure S2. The supplementary S2 analysis is with and without Beer042 as our analysis establish that the Guinness yeast and Beer042 are closely related. We have used the <80% SNPs from a defined geographical location so that our data set is in keeping with other previous published work (Gallone *et al.*, 2016), and the analysis provides, in our opinion, a comprehensive understanding of the Guinness yeast make up.

As noted in point 4, we have included the population structure analysis in our resubmission (figure 1.C), and thank the reviewer for their comment as this omission in the original, manuscript submission was an error.

6. State clearly that the phylogeny in Fig.1 suggests that the closest relatives of the Guinness yeasts are the mosaics/hybrids BE42 (Belgian lager yeast) and BE58 (English beer) (from Gallone et al 2016). The origin of the Guinness yeasts needs to be discussed in more depth. One interpretation is that a single hybridization occurred in the ancestor of BE42/BE58/Guinness isolates, and each lineage then diverged. It is also possible that there were multiple hybridizations. The Structure analysis should help determine if the same parent populations gave rise to all 3 lineages. Phylogenetic analysis of mosaics/hybrids can be difficult.

We thank the reviewer for this comment, as it is a question our Alpaca analysis further supports; relationship of Beer042 and the Guinness yeast (supplementary figure S2). We had hoped to understand the lineage of the BE42/BE58 to the Guinness yeast, and asked Prof Verstappen for more information about these yeast. Our thought process was to then follow up historical information with further genotypic work but unfortunately Prof Verstappen is unable to collaborate or discuss his work with other brewers due to the AB Inbev funding stipulation. This is something we hope to revisit in the future but we do not feel that we can conclude the story successfully without the historical information for context.

7. The discussion of the maltose/maltotriose gene loci is confusing. MAL loci are associated with utilization of maltose (for maltotriose, see next point). *S. cerevisiae* genomes differ in

the number of MAL loci that they contain, and Beer1 isolates usually have many copies. It should be possible to determine the number and location of the loci in the Guinness IDS1 isolate from the MinION assembly. Please include this in a diagram/figure, which will greatly clarify the description. You can then determine if any of the other Guinness isolates have more/fewer loci by mapping the Illumina reads on the MinION assembly. For example, it is very unlikely that MAL63 has moved to Chr XVI. It is more likely that there is an ancestral maltose transporter on Chr XVI (like there is the reference genome) and that MAL63 has been lost.

The *MAL6* operon warrants further investigation as we also found the arrangement of the genes to be unusual, our conclusions are tentative until we have done further work which we expect to do in a follow up paper on the observation. Our MinION assessment of IDS1 and the corresponding scaffolding of the other Guinness yeast suggests that the arrangement of the *MAL6* operon, *MAL61* and *MAL62* on VIII and *MAL63* mapped to the sub telomeric region of chromosome XVI, is replicated throughout all of the Guinness yeast. We assessed the effects of ploidy and our observations were repeated through the sequencing data. Through our assessment we were unable to locate *MAL2* and *MAL4*. The sequencing data (supplementary figure S3) has a homozygous stop codon at position 1075820 in *MAL11* suggesting that this gene is potentially likely to be non-functioning although the occurrence of this stop codon is present within 145 of the 176 yeast assessed in this study; please see line 500. The reported frameshift mutations in *MAL31* are heterozygous and do not lead to a loss in function. Sugar analysis at the end of the fermentations, figure 4, establishes that all the wort sugars are utilised by the Guinness yeast during fermentations. The *MAL63* arrangement conclusions remain tentative until further work is done which we hope to do with a follow up paper.

8. Most MAL loci encode maltose transporters. However, the MAL11 transporter (also called AGT1) at MAL1 is different. This also transports maltotriose. A frameshift in MAL11 would prevent transport of maltotriose, whereas maltose would still be transported if other MAL transporter genes are intact (line 463). Frameshifts in MAL11/AGT1 are indeed unusual in the “British ales” lineage. Are the frameshifts in the 4 Guinness strains homozygous? You should determine which of the Guinness strains can utilize maltotriose and which can utilize maltose, and see if this correlates with the frameshift in MAL11.

We have reviewed the data in light of the referee’s comments, which we thank them for. We can confirm that the *MAL11* frame shift mutations described in the paper’s original submission were heterozygous, however following a review of the data we determined that there was a homozygous premature stop codon present at position 1075820 in all of the Guinness yeast, and 145 of the 176 yeast assessed in our study. The *MAL31* and *MAL61* genes encode maltotriose transporters (Alves *et al.*, 2008; Day *et al.*, 2002). An additional blast search of the Guinness yeast *MAL61* confirmed that the sequence shared a 99% (1837/1842 bp) similarity of a previously described maltotriose utilising gene MTT1 (Dietvorst *et al.*, 2005). Additionally, the lack of maltotriose at the end of fermentation (figure 2) further supports our contention that the Guinness yeast utilise maltotriose.

9. The discussion of off flavors (4-vinyl guaiacol) is interesting. Most Beer 1 yeasts (and all “British ales” yeasts) have frameshifts in FDC1 and PAD1 required to make 4VG (Gallone et al 2016, Goncalves et al 2016, Saada et al 2022), as noted in the discussion. One question is therefore where did the intact genes in the Guinness isolates come from? Were they retained from the ancestor of “British ales” yeasts, and acquired frameshifts in every lineage except for the Guinness isolates, and the wheat beers? My interpretation of supplementary material in Gallone et al is that in BE42 and BE58 (closest relatives of the Guinness yeasts) FDC1 and PAD1 do contain loss-of-function mutations, though in one isolate there are heterozygous mutations. This would suggest that FDC1/PAD1 may have been fully functional in the immediate ancestor of BE42/BE58/Guinness. The functional copies could have originated from a Belgian/German parent, where some isolates have intact FDC1 and PAD1 genes (Saada et al. 2022). This is probably worth discussing. Can you indicate the position of the wheat beer isolates on your phylogeny?

We didn't comment upon BE42 and BE58 for the reasons discussed in reviewers question 6. The wheat beer isolates are highlighted in Figure 1.

10. I can't find a reference to availability of the raw data, though I may have missed it in one of the online documents. The raw data must be available, and the MinION assembly and the annotation must also be provided (if they were available I could look at the MAL loci for example). The custom script (line 222) should also be provided.

The data is now uploaded and the reference included in the manuscript. Line 1169

Illumina and Nanopore (basecalled, demultiplexed) reads for all sequenced samples in this manuscript are deposited in the European Nucleotide Archive (ENA) under the project accession PRJEB62101.

Minor -easily addressed

11. The genome data from Smithwicks, Macardles etc. is underused and little background is provided. Could more information about Perry, Cherry, Smithwicks, McArdles etc. breweries be provided? Do these isolates have mutations in MAL1, FDC1, PAD1 etc? Did one of these lineages contribute to the mosaic structure of the Guinness strains?

We are planning to publish a follow up paper detailing the different phenotypes of the other non Guinness historical Irish brewing yeast. All of the non Guinness breweries and Guinness were members of the Irish Ale Breweries association (estd 1962). The association shared technical information and yeast, so the different lineages established in this first paper were a surprise to us. In addition our phenotypic assessment of the non-Guinness Irish historical yeast suggests that there are phenotypes within the Irish population of the 'British' grouping that are different from the other British yeast.

12. The methods for how the phylogeny was constructed are not provided. A phylogeny

cannot be constructed using SnpEff (line 364). The legend of Fig. 1 refers to Maximum Likelihood, but more information is required.

The omission of the relevant information was incorrect and we thank the reviewer for their comment in highlighting this inaccuracy. We have now amended figure 1 and in the text stating which programmes have been used and the relevant scientific procedure; lines 211-214.

13. One note on using Flye for assembly – this can introduce errors, especially with polyploid eukaryotic genomes. Canu (PMID: 28298431) is often a better choice. You may even have enough raw reads to generate a phased assembly, even for parts of the genome. This would help to determine the origins of the parents of the Guinness strains. (This is a suggestion, not a requirement).

We thank the reviewer for their comment and will explore the use of this programme in potential future works

14. Please clarify the number of strains used, which varies from 19 to 22, depending on whether “IDS1” and “IDS2” etc are assumed to be the same or not (e.g. line 116). Make it clear in Table 1 that “Park Royal” are Guinness isolates.

It was customary for Guinness not to use a completely pure yeast strain, instead they pooled the yeast that were genotypically and phenotypically very similar. This was at the time the accepted procedure for ale and stout brewers with many ale yeast cultures being a mix of yeast. This is contrary to lager brewers who have chosen to use a pure yeast strain of *S. pastorianus*. As an industry we have moved to the pure yeast strain, but due to the historical legacy of the Guinness yeast within the mater culture there are these different variations. This explains the differences in IDS1 and IDS2, subsequently we feel that these yeast should be considered as separate and have been included in the paper as such. We have amended table 1 so that the reader finds it easier to follow.

15. Aneuploidy (Figure 3): there is a difference in the copy number of Chr X in IDS1 and IDS2, which should be commented. You may wish to check Chr XV in FES2. It is surprising that part of this is colored black.

We thank the reviewer and have rechecked our data, the observations of the CNV reported in the paper are correct. We have made a comment in line 453 discussing the different CNV in IDS1 and IDS2.

16. CNVs: did you account for aneuploidy when calculating copy number? BAT2 is on Chr X, and there are extra copies of Chr X in PR1960. The section on CNV and gene content (from around line 504) is somewhat long and speculative.

The analysis in CNV is calculated accounting for the aneuploidy of the yeast. The analysis on the BAT2 was re-evaluated as requested by the reviewer and have confirmed that our original observations are correct. We have amended the manuscript, lines 557– 561.

17. I can't follow the description of 1981 IDS in line 695.

Have amended please see line 748.

18. Which “latter report” is referred to on line 757?

Have included the reference omitting the ‘latter report’ statement. Line 838

19. Line 764: all yeast share the same genetic code??? Do you mean all the relevant strains have the same gene content?

Have amended, line 845.

reference

1. Peris, D., Ubbelohde, E.J., Kuang, M.C., Kominek, J., Langdon, Q.K., Adams, M., Koshalek, J.A., Hulfachor, A.B., Opulente, D.A., Hall, D.J. and Hyma, K., 2023. Macroevolutionary diversity of traits and genomes in the model yeast genus *Saccharomyces*. *Nature Communications*, 14(1), p.690.
2. Peter, J., De Chiara, M., Friedrich, A., Yue, J.X., Pflieger, D., Bergström, A., Sigwalt, A., Barre, B., Freel, K., Llored, A. and Cruaud, C. Genome evolution across 1,011 *Saccharomyces cerevisiae* isolates. *Nature*, 556(7701), pp.339-344 (2018).
3. Gallone, B., Steensels, J., Prahl, T., Soriaga, L., Saels, V., Herrera-Malaver, B., Merlevede, A., Roncoroni, M., Voordeckers, K., Miraglia, L. and Teiling, C. Domestication and divergence of *Saccharomyces cerevisiae* beer yeasts. *Cell*, 166(6), pp.1397-1410 (2016).
4. Abou Saada, O., Tsouris, A., Large, C., Friedrich, A., Dunham, M.J. and Schacherer, J., 2022. Phased polyploid genomes provide deeper insight into the multiple origins of domesticated *Saccharomyces cerevisiae* beer yeasts. *Current Biology*, 32(6), pp.1350-1361.
5. Fay, J.C., Liu, P., Ong, G.T., Dunham, M.J., Cromie, G.A., Jeffery, E.W., Ludlow, C.L. and Dudley, A.M., 2019. A polyploid admixed origin of beer yeasts derived from European and Asian wine populations. *PLoS biology*, 17(3), p.e3000147.
6. Gallone, B., Steensels, J., Mertens, S., Dzialo, M.C., Gordon, J.L., Wauters, R., Theßeling, F.A., Bellinazzo, F., Saels, V., Herrera-Malaver, B. and Prahl, T., 2019. Interspecific hybridization facilitates niche adaptation in beer yeast. *Nature Ecology & Evolution*, 3(11), pp.1562-1575.
7. Alves Jr, S.L., Herbets, R.A., Hollatz, C., Trichez, D., Miletti, L.C., de Araujo, P.S. and Stambuk, B.U., 2008. Molecular analysis of maltotriose active transport and

- fermentation by *Saccharomyces cerevisiae* reveals a determinant role for the AGT1 permease. *Applied and environmental microbiology*, 74(5), pp.1494-1501.
8. Day, R.E., Rogers, P.J., Dawes, I.W. and Higgins, V.J., 2002. Molecular analysis of maltotriose transport and utilization by *Saccharomyces cerevisiae*. *Applied and environmental microbiology*, 68(11), pp.5326-5335.
 9. Dietvorst, J., Londesborough, J. and Steensma, H.Y., 2005. Maltotriose utilization in lager yeast strains: MTT1 encodes a maltotriose transporter. *Yeast*, 22(10), pp.775-788.

** See the Nature Portfolio author and referees' website at www.nature.com/authors for information about policies, services and author benefits

Communications Biology is committed to improving transparency in authorship. As part of our efforts in this direction, we are now requesting that all authors identified as 'corresponding author' create and link their Open Researcher and Contributor Identifier (ORCID) with their account on the Manuscript Tracking System prior to acceptance. ORCID helps the scientific community achieve unambiguous attribution of all scholarly contributions. You can create and link your ORCID from the home page of the Manuscript Tracking System by clicking on 'Modify my Springer Nature account' and following the instructions in the link below. Please also inform all co-authors that they can add their ORCIDs to their accounts and that they must do so prior to acceptance.

If you experience problems in linking your ORCID, please contact the Platform Support Helpdesk.

This email has been sent through the Springer Nature Tracking System NY-610A-NPG&MTS

Confidentiality Statement:

This e-mail is confidential and subject to copyright. Any unauthorised use or disclosure of its contents is prohibited. If you have received this email in error please notify our Manuscript Tracking System Helpdesk team at <http://platformsupport.nature.com>.

Details of the confidentiality and pre-publicity policy may be found here <http://www.nature.com/authors/policies/confidentiality.html>

Privacy Policy | Update Profile

REVIEWERS' COMMENTS:

Reviewer #3 (Remarks to the Author):

I like the discussion a lot now with the relationship with Beer042 properly discussed. I am satisfied with the final revision.